

# Bayesian geological and geophysical data fusion for the construction and uncertainty quantification of 3D geological models

Hugo K. H. Olierook[1], Richard Scalzo[2], David Kohn[3], Rohitash Chandra[2,4], Ehsan Farahbakhsh[2,4], Gregory Houseman[3], Chris Clark[1], Steven M. Reddy[1], R. Dietmar Müller[4]

[1]School of Earth and Planetary Sciences, Curtin University, GPO Box U1987, Perth, WA 6845, Australia
[2]Centre for Translational Data Science, University of Sydney, NSW 2006 Sydney, Australia
[3]Sydney Informatics Hub, University of Sydney, NSW 2006 Sydney, Australia
[4]EarthByte Group, School of Geosciences, University of Sydney, NSW 2006 Sydney, Australia

*Correspondence to*: Hugo K. H. Olierook (hugo.olierook@curtin.edu.au)

**Abstract.** Traditional approaches to develop 3D geological models employ a mix of quantitative and qualitative scientific techniques, which do not fully provide quantification of uncertainty in the constructed models and fail to optimally weight geological field observations against constraints from geophysical data. Here, we demonstrate a Bayesian methodology to fuse geological field observations with aeromagnetic and gravity data to build robust 3D models in a 13.5 × 13.5 km region of the Gascoyne Province, Western Australia. Our approach is validated by comparing model results to independently-constrained

geological maps and cross-sections produced by the Geological Survey of Western Australia. By fusing geological field data with magnetics and gravity surveys, we show that at 89% of the modelled region has >95% certainty. The boundaries between geological units are characterized by narrow regions with <95% certainty, which are typically 400–1000 m wide at the Earth's surface and 500–2000 m wide at depth. Beyond ~4 km depth, the model requires drill hole data and geophysical survey data with longer wavelengths (e.g., active seismic) to constrain the deeper subsurface. Our results show that surface geological

observations fused with geophysical survey data yield robust 3D geological models with narrow uncertainty regions at the surface and shallow subsurface, which will be especially valuable for mineral exploration and the development of 3D geological models under cover.

## 1    Introduction

Surface mapping and subsurface interpretation of geological units is one of the tenets of the geological and geophysical community. Geological units and their associated boundaries may share similar geological histories or may be juxtaposed to one another via structural discontinuities such as faults or suture zones. Accurately positioning geological units and their boundaries is fundamental for, and not limited to, constraining plate reconstructions (Cawood and Korsch, 2008; Merdith et al., 2017), defining stratigraphy (Gradstein et al., 2012), and successful mineral and petroleum exploration (Dentith and

Mudge, 2014; Selley and Sonnenberg, 2014). In order to conceive a comprehensive model of both the surface and subsurface geology, a combination of geological mapping, geophysical interpretation, sample analysis and prior knowledge are used.



Although all of these ingredients are important, contemporary workflows to incorporate them all tend to develop at best only a handful of possibly biased solutions that either neglect, or incompletely account for, the uncertainty associated with geological or geophysical interpretation, as well as the knowledge of how far to extrapolate data derived from sample analysis. This shortcoming of traditional geological model-building is exacerbated in regions with thick sedimentary or regolith 'cover'

because this poses a significant impediment to understanding the nature of the subsurface.

To develop robust geological models, it is important to quantify the uncertainty on the position and configuration of geological units. Previous work addressing the uncertainty problem included employing fuzzy logic and information entropy approaches to build semi-quantitative 3D geological models (Abedi and Norouzi, 2012; Joly et al., 2012; Wellmann and Regenauer-Lieb,

2012). However, these approaches still require a significant degree of human decision making into how to fuse disparate geoscientific datasets. Other approaches characterize uncertainty by generating ensembles of 3D models through perturbations of a set of underlying descriptive geometric parameters (de la Varga et al., 2018; Giraud et al., in review; Lindsay et al., 2013; Pakyuz-Charrier et al., 2018). However, these approaches still largely elide the question of how the joint distribution of such parameters is meant to be derived. A fully quantitative and informative 3D geological model will fuse all available constraints

in a probabilistically rigorous fashion. Bayesian inference provides a suitable framework for doing this by using Markov chain Monte Carlo (MCMC) sampling methods for estimation and uncertainty quantification of free parameters. Previous studies using Bayesian inference in the geosciences have focused on (i) geophysical joint inversions (Bosch et al., 2006; Giraud et al., 2017; Shen et al., 2013), (ii) fluid flow through permeable reservoirs for groundwater, hydrocarbon or carbon dioxide storage applications (Oladyshkin et al., 2013; Refsgaard et al., 2012; Seifert et al., 2012; Ye et al., 2010), and (iii) geomorphologic

and climate evolution (Chandra et al., 2018; Hapke and Plant, 2010; Pall et al., 2018). However, there is still a paucity of work in fusing solid Earth geological observations and geophysical data in a Bayesian framework to develop robust 3D geological models.

The Obsidian software package provides a workflow to fuse disparate geological and geophysical data within a Bayesian

framework (McCalman et al., 2014; Reid et al., 2013). Obsidian features a parameterized world model of the 3D structure and physical properties of geological formations. Here, the free parameters are estimated by MCMC with the sampling taking into account both the estimated prior probability of the existence of any particular formation, and the likelihood of that configuration producing all available geophysical survey data in the modelled region. Obsidian was originally designed for deep (~1–5 km) geothermal energy applications in sedimentary basins, which includes the ability to fuse airborne or surface geophysical

surveys (e.g., aeromagnetics, gravity, magnetotellurics, temperature) with laterally-sparse geophysical drill hole data (e.g., geological unit depths, bore hole temperature, density) and drill hole geological units as prior points. One useful addition to the current features of Obsidian would be the integration of geological and geophysical field observations made on the Earth's surface, which are vital for surface and near-surface applications (< 1 km). Such applications include the mineral exploration sector but also extend to any igneous or metamorphic ('hard-rock') terranes. Unlike sparse drill hole data, surface geological



and geophysical observations provide high-resolution lateral constraints on 2D surface geological models that, together with geophysical survey data, permit the development of robust 3D geological models.

In this contribution, we extend the Obsidian software package to enable coupling of (i) airborne magnetic and gravity survey data (using petrophysical priors derived from surface samples) with (ii) geological field observations that inform the configuration of geological units at discrete points on the Earth's surface. We demonstrate the validity of our techniques by building models of a data-rich, $13.5 \times 13.5$ km subsection of the Gascoyne Province, Western Australia, and comparing the model results to surface geological maps produced by the Geological Survey of Western Australia (GSWA) and subsurface interpretations (Johnson et al., 2013). The chosen study area is particularly suitable as it also exhibits a significant portion of recent sedimentary and regolith cover, which makes certain areas inaccessible for recording geological surface observations directly but possible to infer using Bayesian techniques. We anticipate that this framework provides a foundation for future applications in igneous and metamorphic terranes, particularly in the mineral exploration sector and especially in exploration under cover (McFadden et al., 2012).

## 2 Background

### 2.1 Geological setting of the Gascoyne Province

The Gascoyne Province, and the wider Capricorn Orogen, record the protracted amalgamation of the West Australian Craton and subsequent intracontinental tectonothermal activity (Fig. 1, Fig. 2). Two main events are thought to contribute to forming the West Australian Craton. First, the ca. 2195–2145 Ma Ophthalmia Orogeny sutured the Glenburgh Terrane, comprised of the Halfway Gneiss, to the Pilbara Craton (Krapež et al., 2017; Rasmussen, 2005). The deposition of the Moogie Metamorphics was associated with the Ophthalmia Orogeny, deposited into a foreland basin that formed in a response to the Glenburgh-Pilbara collision (Johnson et al., 2013). Second, the ca. 2005–1950 Ma Glenburgh Orogeny then amalgamated the combined Pilbara Craton-Glenburgh Terrane with the Yilgarn Craton to form the West Australian Craton (Johnson et al., 2013; Olierook et al., 2018). The Glenburgh Orogeny was associated with two major Andean-type granitoid formations, the Dalgaringa and Bertibubba Supersuites, and several subduction-related basins (Johnson et al., 2011; Olierook et al., 2018). After unification, the Capricorn Orogen experienced at least five intracontinental tectonomagmatic events, each decreasing in severity of tectonic character and magmatism (Johnson et al., 2017). The first two events, the 1830–1780 Ma Capricorn Orogeny and 1680–1620 Ma Mangaroon Orogeny, were both associated with significant granitoid magmatism of the Moorarie and Durlacher Supersuites, respectively (Sheppard et al., 2010a; Sheppard et al., 2005). Deposition of the Leake Springs Metamorphics was also concurrent with the early stages of the Capricorn Orogeny. Later events were predominantly amagmatic but were still associated with up to amphibolite-facies metamorphism and hydrothermal activity (Korhonen et al., 2017; Sheppard et al., 2007). Both suturing and intracontinental tectonic events have developed a pervasive east–west striking structural fabric in the





Gascoyne Province that has compartmentalized the region into several geological zones that share tectonic characteristics (Sheppard et al., 2010a). In the south, zone and formation boundaries trend NE–SW whereas major structures are oriented NW–SE in the north, yielding a wedge-shaped geometry for the Gascoyne Province (Fig. 1, Fig. 2). Compared to the rest of the Capricorn Orogen, the Gascoyne Province is relatively well exposed but there are still significant areas covered by recent
regolith and sediment that hamper mineral exploration.

## 2.2    Bayesian inversion and inference

Inverse problems aim to recover the causal factors that produced a set of observations (Mosegaard and Tarantola, 1995; Sambridge and Mosegaard, 2002). For geological and geophysical applications, the objective of inverse problems is to
recover the subsurface properties such as density and magnetic susceptibility from surface-based geophysical survey measurements of gravity and magnetic field strength. However, geophysical surveys cannot yield unique solutions of the subsurface petrophysical properties. Thus, there are an infinite number of subsurface petrophysical measurement configurations that would produce the same survey readings (Sambridge, 1999; Sambridge and Mosegaard, 2002). Given that there is no single unique solution, there is no reason to prefer one model over another without introducing constraints on
what form the model should take (Parker, 1977; Sambridge and Mosegaard, 2002). Ways to introduce such constraints include regularization (Giraud et al., in review) but this technique fails to acknowledge alternative scenarios. A probabilistic interpretation casts the problem in terms of maximizing  the likelihood or *a posteriori* probability of the model, but the fact remains that multiple qualitatively different solutions may be equivalently accessible to the data (Mosegaard and Vestergaard, 1991; Rocca et al., 2009).

An alternative approach is to use sampling over all possible models in a probabilistic Bayesian context, which provides a more systematic approach towards uncertainty quantification and the incorporation of prior constraints. Bayesian inference can be applied to parametric (Shafer, 1982) and nonparametric models (Hjort et al., 2010). Bayesian methods have become more popular in geophysics in the past few decades (Malinverno, 2002; Mosegaard and Tarantola, 1995; Sambridge and
Mosegaard, 2002; Sambridge and Compston, 1994). The strategy of obtaining free parameters that fit the data in geophysical models have shifted from optimization (Sen and Stoffa, 2013) to inference that addresses uncertainty quantification given that the models provide an approximation of geophysical processes (Gallagher et al., 2009). The Bayesian framework converts a deterministic model into a probabilistic one by using probability distributions to represent the free parameters rather than using optimal or single-point estimates. This takes into account observed data and prior information (priors)
about the model parameters (Oldenburg 2005). The prior information about model parameters is presented in the form of a probability distribution (e.g., density and magnetic susceptibilities; Backus, 1970). The likelihood function evaluates the quality of the proposal for the free parameters by taking into account the forward model output, given data and noise in model outputs (Guillaume et al., 2013; Kaipio and Somersalo, 2006). The priors and likelihood functions multiply to give



solutions to the inverse problem, which is referred to as the posterior distribution (Kaipio and Somersalo, 2006; Oldenburg and Li, 2005).

The approximation of the posterior distribution by MCMC methods is computationally expensive since thousands of model
evaluations are required to iteratively sample the posterior distribution (Sen and Stoffa, 1996; Tarantola and Valette, 1982).
To do this, samples (proposals) are drawn from a target distribution by constructing a Markov chain that, after a number of
steps, converges to the desired distribution as its reaches equilibrium (Hastings, 1970; Kass et al., 1998; Metropolis et al.,
1953; Raftery and Lewis, 1996; van Ravenzwaaij et al., 2018). Convergence criteria determine when to stop sampling that, for
example, could be a predetermined number of samples or an assessment of the behaviour of the likelihood function. However,
for complex and large-scale 3D inversion problems, convergence can be challenging due to the large number of free parameters
that need to be sampled effectively in limited computation time (i.e., high dimensionality; Sen and Stoffa, 1996).

For multimodal posteriors, the Markov chain can become trapped in a single mode and cannot fully explore the posterior
distribution, making sampling much less efficient or causing the chain to converge to the wrong distribution. Parallel tempering
(PT) is a sophisticated MCMC method that aims to increase the efficiency of the exploration of multimodal posterior
distributions (Geyer, 1993; Hukushima and Nemoto, 1996; Sambridge, 2013). Parallel tempering uses a number of replicas of
the original sampling method, where the replicas are created at different 'temperatures' (Brooks et al., 2011; Earl and Deem,
2005) by rescaling the likelihood probability density function (Sambridge, 2013). High temperature replicas sample a smoother
(flatter) version of the likelihood function in order to 'escape' from local minima and provide global exploration features.
Conversely, low temperature replicas provide local exploration capabilities. Hence, with parallel tempering, there is a delicate
balance between global and local exploration (Earl and Deem, 2005; Sambridge, 2013). During sampling, the replicas are able
to exchange their configurations, typically between neighbouring replicas via the Metropolis–Hastings proposal (Sambridge,
2013). Ultimately, this improves the mixing of Markov chain (i.e. thorough exploration of space and convergence to the target
distribution) and efficiency of convergence (Sambridge, 2013).

## 2.3     Obsidian software package for joint geophysical inversion

The Obsidian software package was originally designed for geothermal exploration in sedimentary basins. Here, we provide a
brief overview of the salient features that are important for our inverse problem (Fig. 3). For a detailed background of Obsidian,
the reader is referred to Ramos et al. (2012), Reid et al. (2013), McCalman et al. (2014), Beardsmore et al. (2016) and Scalzo
et al. (in review).

The modelled area in Obsidian is parameterized as a series of discrete layers, each with its own spatially constant rock
properties, separated by smooth geological boundaries (Fig. 3a, b; Beardsmore et al., 2016; McCalman et al., 2014). Each





geological layer boundary is a 2D Gaussian process regression against a set of user-defined control points ($\alpha_n$ in Fig. 3a) that specify the subsurface depth of the boundary at given surface locations. The layer boundaries are indexed in a strict order of increasing depth in the subsurface but are permitted to cross. For each layer, the control point depth offsets have a multivariate Gaussian prior with mean zero and a specific covariance between the control points for that layer (Fig. 3a). For each layer, our
inverse problem comprises: (i) parameters for the offset of the mean depth from the top of that layer at each control point position, and (ii) rock properties for each geophysical survey (e.g., magnetic susceptibility property for aeromagnetic field strength, $\rho_n$ in Fig. 3a).

The Gaussian processes that interpolate the layer boundaries across the lateral extent of the modelled volume use a radial basis
function kernel with a mean function that varies by the layer and fixed x and y correlation lengths (Fig. 3c; Beardsmore et al., 2016; McCalman et al., 2014). These correlation lengths are fixed to the spacing between control point locations. The rock properties (e.g., magnetic susceptibility) for each layer are independent of the control points and have a multivariate Gaussian prior. For each geophysical survey (e.g., aeromagnetics), the likelihood is Gaussian but the variance of the 'noise' is assumed to follow an inverse gamma distribution with user-specified hyper parameters that define the shape of the distribution. When
the noise hyper parameters are integrated via sampling, the resulting likelihood when marginalized over an inverse gamma prior is a Student's t with $2\alpha$ degrees of freedom and a scale of $\beta/\alpha$ for the observations.

The PTMCMC algorithm embedded in the Obsidian software package optimizes the mean acceptance rate of the swaps between the chains in adjacent temperatures by continuously adapting the scale parameter of the proposal distribution (i.e., the
estimated covariance of the target distribution at each temperature) during the simulation (Beardsmore et al., 2016; Miasojedow et al., 2013). This continuous adaptation – known as adaptive parallel tempering MCMC – allows the algorithm to learn the progressive adjustment of step size used for proposals within each chain as well as the temperature ladder used to sample across chains (Andrieu and Thoms, 2008). The maximum allowed change to any chain property diminishes over time, inversely proportional to the number of samples to ensure convergence (Andrieu and Thoms, 2008).

## 3    Materials and Methods

### 3.1    World parameterization

The construction of the parameterized world model in the chosen $13.5 \times 13.5$ km area in the Gascoyne Province involves three types of data: a hierarchical construction of layers using available 2D seismic data interpretation (Fig. 1b; Johnson et al., 2013)
and two types of point-based measurements — magnetic and density data — on hand samples from Aitken et al. (2014). A two-dimensional seismic survey was conducted in 2011 (Johnson et al., 2013). The surface position of the seismic line is immediately to the west of our study area and cross-cuts the same geological units (Fig. 1a). The seismic interpretation of





Johnson et al. (2013), aided by geochronological data, directly informs that the ordering of layers in the world model, from oldest to youngest, are: (i) the ca. 2550–2430 Ma Halfway Gneiss (Johnson et al., 2017), (ii) ca. 2210–2150 Ma Moogie Metamorphics (Martin and Morris, 2010), (iii) ca. 1840–1810 Ma Leake Spring Metamorphics and ca. 1830–1780 Ma Moorarie Supersuite (Johnson et al., 2011; Sheppard et al., 2010b), (iv) ca. 1690–1660 Ma Durlacher Supersuite (Piechocka

et al., 2017; Sheppard et al., 2005), and (v) ca. 995–900 Ma Thirty Three Supersuite (Piechocka et al., 2017; Sheppard et al., 2010b). In the chosen 13.5 × 13.5 km area, only the Halfway Gneiss and Durlacher Supersuite are areally significant, comprising ~59% and ~35% of the interpreted area by GSWA (Fig. 2). Other areally-minor units in the 13.5 × 13.5 km area include the Leake Spring Metamorphics (~3%), Moogie Metamorphics (~2%), Moorarie Supersuite (<1%) and Thirty Three Supersuite (<1%).

### 3.2    Petrophysical data

Two types of petrophysical data, magnetic susceptibility and density, were collected on hand samples (Aitken et al., 2014) to link to aeromagnetic and gravity data, respectively (Fig. 4). A relatively low number of magnetic ($n = 104$) and density ($n = 103$) samples across the entire Gascoyne Province were available (Aitken et al., 2014). Thus, the sample mean and covariance

of magnetic and density measurements for each geological unit were used to inform a multivariate Gaussian prior, acknowledging that spatial differences in magnetic and density distributions cannot be captured in this contribution.

### 3.3    Geophysical and geological data

Two geophysical survey data are employed, namely aeromagnetic (Fig. 5a) and gravity data (Fig. 5b), that are forward-

modelled to correspond to sample-based magnetic and density data, respectively. These types of geophysical surveys were already available for Bayesian fusion in the Obsidian framework. In addition to the geophysical data, field-based geological units observations are incorporated into Obsidian (Fig. 5c).

Aeromagnetic data in the study area (Fig. 5a) utilized a subsection of the 1995/96 Bangemall Survey directed by GSWA from

latitudes 23.5–26.0° S and longitudes 115.0–120.0° E (Geological Survey of Western Australia, 1996). The Bangemall aeromagnetic data were flown at a 7.5 m sample interval, 500 m flight line spacing and a mean terrain clearance of 60 m. The final magnetic intensity in nT has the following corrections from the raw data: (i) the 1990 IGRF model removed and a base value of 54940 nT added, (ii) diurnal correction applied, with a base value of 55220 nT, (iii) parallax correction of 0.4 fiducial applied, (iv) levelled using tie line information, and (v) tie lines force levelled to flight lines. The original horizontal datum of

the Bangemall Survey was the AGD84, projected using AMG zone 50, but this was converted to the WGS84, zone 50 S for each data point. The flight path vector data were explicitly favoured over the post-processed raster data to avoid introducing correlations.



Gravity anomaly data in the study area (Fig. 5b) were derived from the 2010 Gascoyne North and Gascoyne South surveys directed by GSWA from latitudes 23.5–26.0° S and longitudes 115.2–118.5 ° E (Mathews and Jecks, 2010). The Gascoyne North and South gravity data were acquired at a ground-based nominal station spacing of 2500 m in a square grid configuration.

The final complete spherical cap Bouguer anomaly in $\mu ms^{-2}$ had the following corrections from the raw data: (i) correction of remanent drift, typically less than 0.05 $\mu ms^{-2}$ $hr^{-1}$, (ii) computation of Bouguer anomaly using a modified spreadsheet developed by M. Bacchin of Geoscience Australia, (iii) spherical cap Bouguer anomaly computation relative to the Australian Absolute Gravity Datum 2007, and (iv) terrain correction using the AUSGEOID09 vertical coordinate reference frame. The original horizontal datum was GDA 94, which is equivalent to WGS84. The located surface point-based data were explicitly

favoured over the post-processed raster data. Very finely, regularly-sampled raster data are often preferred in order to apply fast Fourier transform inversion techniques. However, resampling or interpolating non-gridded data onto such a grid results in correlations between the gridded data points, which can lead to biases and incorrect results in probabilistic inversions if not explicitly accounted for (Scalzo et al., in review).

Surface geological unit observations were acquired from the Western Australian Rocks (WAROX) database, available from GSWA (Fig. 2, Fig. 5c). Each spatially-referenced sample point records an observation of rock type coupled to an interpreted geological unit. Designation of a particular geological unit is informed from petrographic, geochemical and geochronological knowledge obtained on a subset of WAROX data. For example, the assignment of a geological unit is near certain where U–Pb crystallization ages are available or where whole-rock major and trace element geochemistry has been collected. In the

chosen $13.5 \times 13.5$ km study area, only one sample has geochronological and geochemical information (Fig. 5c). However, there are >100 age and >500 samples with geochronological and geochemical data, respectively, in the Gascoyne Province from the same geological units that are present in the $13.5 \times 13.5$ study area (e.g., Johnson et al., 2017). All samples with U–Pb ages and/or geochemical data in the Gascoyne Province also have petrographic and/or hand sample descriptions, which can be used to inform the geological unit for geological surface observations where only hand sample or petrographic descriptions

are available. Even though inference of geological units from similar petrographic and hand sample descriptions is relatively robust, it may be in error. Thus, we have accounted for this potential uncertainty in geological field observations (see section 3.4 for further details). All our observations were taken at the surface but could be readily used where geological unit observations could be made in the subsurface (i.e., via drill hole information). The formation that is observed at the surface defines the value of a given field observation.

## 3.4    Likelihood models

We use Gaussian likelihood distributions for gravity and aeromagnetic surveys, which assumes that the residuals from the respective forward models are themselves Gaussian. The mean of each survey's Gaussian likelihood is given by the output of



the respective survey's forward models. The unknown variance of each Gaussian likelihood distribution is drawn from an inverse gamma prior, with survey-specific shape parameters. For both the gravity and magnetism survey data, we set the noise α to 1 and the noise β to 0.2. The inverse gamma priors allow us to incorporate uncertainty for the noise for the survey likelihoods. We assume that the likelihoods are conditionally independent given the world model output (McCalman et al,

5   2014).

We use a binomial likelihood for the field observations and we put a Beta prior probability distribution on the probability of success for the binomial model. The binomial likelihood is defined as:

$$P(k|n,p) = \frac{\Gamma(n+1)}{\Gamma(k+1)\Gamma(n-k+1)} p^k (1-p)^{n-k}$$

where $k$ is the number of successes, $n$ is the number of trials, $p$ is the probability of success for each trial and $\Gamma$ is the gamma function. The left fraction is the binomial coefficient and is more commonly written as $\binom{n}{k}$ but the gamma function representation generalizes to non-integers. With a Beta hierarchical prior on $p$, we can write:

$$P(k|n,\alpha,\beta) = \frac{\Gamma(n+1)}{\Gamma(k+1)\Gamma(n-k+1)} \int_0^1 p^k (1-p)^{n-k} \left[ \frac{p^\alpha (1-p)^\beta}{B(\alpha,\beta)} \right] dp = \frac{\Gamma(n+1)}{\Gamma(k+1)\Gamma(n-k+1)} \frac{\Gamma(k+\alpha)\Gamma(n-k+\beta)}{\Gamma(n+\alpha+\beta)}$$
$$+ \frac{\Gamma(\alpha)\Gamma(\beta)}{\Gamma(\alpha+\beta)}$$

The Beta distribution is a distribution that takes on values between zero and one and has two shape parameters, α and β. For the field observation likelihood distribution, we set the noise α to 20 and the noise β to 1, resulting in a 95% credible lower limit of getting at least 85% correct observations in the field ID dataset. Using a Beta prior results in an analytically tractable likelihood that captures potential overdispersion in the field observations. Overdispersion could be a problem if there is more than one source of variation in the error of the observations, such as distinct geologists interpreting rock formations differently

(Gelman et al., 2013).

### 3.5   Markov chain Monte Carlo sampling

We use PT-MCMC to explore the parameter space, which is limited by our world parameterization. We can use different types of PT-MCMC proposals that to varying degrees must maintain the properties of the Markov chain to ensure convergence to

the target distribution. The original Obsidian implementation used an isotropic Gaussian random walk (iGRW) proposal (Beardsmore et al., 2016; McCalman et al., 2014). To increase the efficiency of our sampler as to draw less correlated and more independent samples, we instead use the preconditioned Crank–Nicolson MCMC proposal, which weights between a Gaussian random walk proposal and a draw from the prior (Cotter et al., 2013; Hu et al., 2017; Rudolf and Sprungk, 2018). Further details of our implementation can be found in Scalzo et al. (in review).



## 3.6    Experiment Design

We run the Obsidian PT-MCMC with six parallel tempering temperature ladders (stacks), where each stack consists of twelve PT-MCMC chains. The likelihood of each chain is raised to the power of a different temperature in the ladder and the lowest temperature chain is the unnormalized likelihood. Samples are only collected for the lowest temperature chain. This setup has

enough chains in each ladder to ensure geometric spacing between temperatures on the ladder, confirmed by empirical examination of the ladder, and has enough stacks to ensure sufficient confidence in the convergence diagnostics, specifically the potential scale reduction factor diagnostic. The experiment was run on an area of 13.5 x 13.5 km for 96 hours to ensure convergence for all rock property and control point parameters.

# 4    Results

## 4.1    Convergence diagnostics

Convergence diagnostics aid in evaluating whether the MCMC sampling has converged (i.e., whether sampling is occurring from the target distribution; Gelman et al., 2013). Convergence of the control point parameters occurred after only 12 hours but the rock property parameters required approximately half of the total 96-hour run time to reach convergence. Several

techniques are listed here to confirm that our model outputs are statistically-valid, including (i) trace plots of the MCMC samples (Fig. 6a, b, Table 1), (ii) autocorrelation times and effective sample size (Fig. 6c, Table 1), (iii) potential scale reduction factor (Table 1), and (iv) Geweke score (Fig. 7).

Trace plots for modelled density and magnetic susceptibility show that: (a) chains initialized at different initial states have

similar posterior densities, and (b) chains mix well, i.e., they sufficiently explore the support of the posterior distributions as determined by the parameters' respective priors (first column in each panel of Fig. 6). The Halfway Gneiss and Durlacher Supersuite have modelled densities of $2.72 \pm 0.12$ and $2.67 \pm 0.12$ g cm$^{-3}$ ($2\sigma$), respectively, and average modelled log$_{10}$ magnetic susceptibilities of $-3.65 \pm 0.57$ and $-2.60 \pm 0.07$ ($2\sigma$), respectively (second column in each panel of Fig. 6). Compared to the Halfway Gneiss, a lower magnetic mean susceptibility and larger variance in magnetic susceptibility for the Durlacher

Supersuite agrees with the prior density and magnetic measurements for both formations (Fig. 4). The lack of difference between the modelled densities of the Halfway Gneiss and Durlacher Supersuite (at $2\sigma$) are also in agreement with the priors (Fig. 4).

There are approximately 1.5 million total samples for each chain. However, samples from MCMC are correlated (third column

in each panel of Fig. 6), which reduces the number of independent samples (i.e., the effective sample size) from the posterior distribution (Gelman et al., 2013). The MCMC autocorrelation is on the order of 1 in 12 to 1 in 14 independent samples per



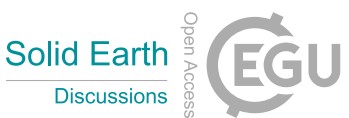

MCMC proposal for rock property parameters and 1 in 1 to 1 in 15 independent samples for control point parameters (Table 1). This means that there are approximately 105,000–129,000 and 100,000–1,000,000 independent samples for the rock property and control point parameters, respectively.

The potential scale reduction factor (PSRF), also known as the Gelman-Rubin statistic or $\hat{R}$ (Gelman and Rubin, 1992), assesses convergence by comparing the variance between means of multiple chains relative to the average of the variance within chains to show how much an estimator of the marginal posterior variance will decrease as the number of samples increases (Brooks and Gelman, 1998; Cowles and Carlin, 1996). If the diagnostic is close to 1 then limited reductions in variance can be made from further sampling and the sampling has likely converged to the target distribution. All of our rock property and control

point parameters have $\hat{R} = 1.02$–$1.03$ and $\hat{R} = 1.00$–$1.06$, respectively (Table 1), which indicate convergence on the basis that the Gelman-Rubin statistic is less than the threshold of 1.10 (Gelman et al., 2013).

   The convergence of the target distribution may also be evaluated using the Geweke score (Geweke, 1991), which is a z-score diagnostic that compares the mean of subsets of samples from the start and end of the MCMC chains. A heuristic for

convergence are Geweke scores between –2 and +2, indicating normality of the difference in means (Cowles and Carlin, 1996). We see convergence of our chains in terms of this diagnostic with Geweke scores of and –1 to +1 and –0.5 to 1.5 for the density and magnetic susceptibility parameters, respectively, except for two chains (chain 2, density for both formations) which show large deviations (Fig. 7).

**4.2     Residuals from forward models**

   Aeromagnetic and gravity models from forward models are broadly comparable to their measured counterparts (Fig. 8a, d). Aeromagnetic models effectively identify the NW–SE strike of magnetic lineaments in the northern half of the modelled volume (Fig. 2, Fig. 8). The NE-trending elongate unit in the southern half of the map, corresponding to the Durlacher Supersuite (Fig. 2), also shows limited discrepancies between modelled and measured data (Fig. 8). Aeromagnetic residuals

display an approximately Gaussian distribution of $0^{+358}_{-317}$ nT (2σ, 21% of the total magnetic range; Fig. 8). Only one region in the northwestern portion of the map has significantly higher magnetic field strength than modelled (Fig. 8).

Modelled gravity effectively identify the long-wavelength, N–S trending structure in the measured data but there are significant positive residuals in the south-west corner and negative residuals in the central-eastern portion of the modelled area (Fig. 2, Fig. 8). Gravity residuals are positively skewed with a residual of $0^{+3.96}_{-2.36}$ mGal (2σ, 20% of the total gravity

range; Fig. 8).





The forward modelled field observations have a posterior probability of success of approximately 80% (Fig. 9a–c). Five of 141 actual Halfway Gneiss observations have been misclassified as belonging to the Durlacher Supersuite, while most of the actual Durlacher Supersuite observations have been misclassified as belonging to Halfway Gneiss (Fig. 9d–f). All the misclassifications occur within 1 km of boundaries between geological units, particularly in the southeast and eastern parts of the 13.5 × 13.5 km area (Fig. 9e).

### 4.3 Probability density of layer locations

Voxelized posterior distributions of the modelled volume reveal a strong probability contrast between regions of high certainty (defined as >95%) at the surface (Fig. 10). Two modelled volume shows that the Durlacher Supersuite occupies the northeastern section of the region and an ellipsoidal inlier towards the southern extent of the 13.5 × 13.5 km map (Fig. 10). The remainder of the map shows Halfway Gneiss. For the NE–SW boundary between the modelled Durlacher Supersuite and Halfway Gneiss, the distance between >95% certainty of Halfway Gneiss and Durlacher Supersuite (i.e., region with <95% confidence) equates to horizontal distances of ~300–1000 m at the surface. The ellipsoidal Durlacher Supersuite inlier is heterogeneously constrained. The horizontal distance for the boundary between the Durlacher Supersuite and Halfway Gneiss (<95%) is relatively tightly constrained along the NW and SW margins (~450 m), moderately constrained along the SE margin (~750–1100 m) and poorly constrained towards the east (up to 2350 m; Fig. 10).

At depth, sub-vertical unit boundaries are maintained as informed by the prior (Figs. 1, 2; seismic interpretation of Johnson et al., 2013). The modelled cross-section yields dips of 85° near the surface, and progressively reducing in inclination to ~72° at 4 km depth. In the cross-section, the horizontal distance between regions of >95% certainty between the Halfway Gneiss or Durlacher Supersuite becomes progressively more diffuse, from ~420 m to 1060 m between the surface and 4 km depth, respectively (Fig. 11). This translates to a percentage decrease in horizontal confidence of ~250%. In other parts of the 3D model, regions of <95% certainty can be as wide as ~2500 m at depth (Fig. 11b).

### 5 Discussion

### 5.1 Validity of 3D models and comparison to geological maps and cross-sections.

The fusion of geological field observations with gravity and magnetic data are valid on a statistical basis, including showing: (i) modelled petrophysical properties comparable with the prior (Figs. 4, 6), (ii) adequate yield of independent samples (Fig. 6), (iii) sufficient exploration of the parameter space ($\hat{R} < 1.1$, Table 1), and (iv) convergence as indicated by Geweke scores between –1 and +1 (Fig. 7). Aeromagnetic, gravity and field observations show mean residuals of ~0 with 2σ tails that are a maximum of ~20% of the total dataset range (Fig. 8c,f). Aeromagnetic residuals are spatially uncorrelated except for a small




region in the NW corner of the 13.5 × 13.5 area (Fig. 8b), indicating that the model captures relevant variation on the length scales of interest. Gravity residuals are systematically positive in the south, west and north, and negative in the east (Fig. 8e). This is primarily a function of a long-wavelength (i.e. deep) gravity response, whereas the model aims to capture the shallow subsurface model (<5 km). Field observation misclassifications are only found within 1 km of geological boundaries. The

discrepancies between field observations and modelled geological units may have resulted from the presence of other geological units (particularly those that are highly magnetic). Additionally, more petrophysical data for the Halfway Gneiss and Durlacher Supersuite may have yielded better priors for the geophysical surveys, which in turn would have corroborated better with the position of geological field observations. Ultimately, the data residuals are sufficiently small to have yielded a reliable model output despite minor discrepancies.

The voxelized posterior distributions of the modelled volume are visually comparable to geological maps and interpreted cross-sections made by GSWA (Fig. 2, Fig. 10). At the surface, the NW–SE striking Chalba Shear Zone boundary between the Halfway Gneiss and Durlacher Supersuite and the ellipsoidal inlier of Durlacher Supersuite are effectively captured in the models, with predominantly <1 km of <95% confidence regions separating >95% certainty domains (Fig. 10). However, there

is an additional ~1 km wide NW–SE spur of Durlacher Supersuite immediately south of the main portion of the Chalba Shear Zone that is not captured in the models. Additionally, a thin sliver of mapped Durlacher Supersuite that encroaches the map in the NW section of the map (at ~7255500 mN) is modelled as Halfway Gneiss (Fig. 10), which explains why an abnormally high magnetic residual is present there (Fig. 8). Both of these discrepancies are probably a result of the aeromagnetic data integrating the magnetic response of Halfway Gneiss at depth. For example, in the GSWA cross-section across the NW–SE

spur of Durlacher Supersuite, this NW–SE spur is interpreted to be underlain by Halfway Gneiss at depths below 2 km (Fig. 2, Fig. 10).

In three-dimensions, the model maintains the sub-vertical to steeply-dipping regions of <95% certainty (i.e., geological boundaries; Fig. 11). This is particularly well viewed in the X–Y cross-section, where the posterior distributions reveal sub-

vertical dips (>85°) that are comparable to the sub-vertical dips measured in the field and propagated into interpreted cross-sections (Fig. 2, Fig. 10). The modelled inclinations at 4 km depth are shallower (72°) than those interpreted by GSWA, which maintain dips of >85° at 4 km (Fig. 2). Seismic interpretation data indicates that the Chalba Shear Zone is dipping at ~65° at 4 km depth (Fig. 2), more comparable to our modelled dips that those interpreted from geological mapping. However, with the lack of drill hole data, it is difficult to know exactly whether the dips obtained from seismic interpretations, geological cross-

section interpretation or modelled posterior distributions are correct. Despite these small discrepancies, the broad architecture of the model maintains the framework inferred from geological maps and cross-sections. Importantly, our method is the only technique that provides a range of solutions and quantitatively accounts for all the input assumptions.





Another important output is that the modelled posterior distributions reveal that the Durlacher Supersuite is definitively separated into two domains, one NE of the Chalba Shear Zone and the other as an ellipsoidal inlier, with a ~2.5 km-wide spur of >95% confidence Halfway Gneiss separating the two regions (Fig. 10). This model output is corroborated at the surface by geological mapping across the region (Fig. 2) but it was difficult to know whether this spur of Halfway Gneiss between the two Durlacher Supersuite domains continued at depth or was truncated in the near subsurface. Our results indicate that the spur of Halfway Gneiss continues until at least 4 km as assumed from geological mapping. This important contribution shows that small geological volumes on the scale of a few km can be resolved accurately and will be important when this modelling output is up-scaled to larger regions.

## 5.2    Implications and limitations for quantification of uncertainty in 3D geological models

To develop robust 3D geological models, fusion of geological and geophysical data in a fully probabilistic (Bayesian) method are vital for pure (e.g., plate reconstructions) and applied geological problems (e.g., mineral exploration). At the surface and near-surface (1 km), our model results are highly similar to independently-constrained geological maps and interpreted cross-sections (Fig. 2, Fig. 10), which are useful for mineral exploration applications that rarely exceed economic deposit depths of 1 km (McFadden et al., 2012). At the surface, distances between domains of >95% confidence rarely exceed 1 km, although the eastern part of the ellipsoidal Durlacher Supersuite inlier are as wide as ~2350 m. The minimum horizontal distance of <95% uncertainty at geological boundaries appears is ~400 m, which appears to be inherently linked to the line spacing of the aeromagnetic survey. Given that the gravity survey and geological field observations are far more widely spaced, and therefore have less control on the model outputs, the aeromagnetic data distribution is probably the dominant control on the width of uncertain regions. So, if higher resolution geophysical surveys and/or geological field observations are acquired, the model can then become more precise. For geological mapping applications (particularly in the mineral exploration sector), geological mapping in these uncertain regions (if outcrop is available) and/or high-resolution geophysical surveys across these small regions of uncertainty provide targeted and cost-effective methods of yielding better 3D geological models. Where such regions are under cover and drilling is required to establish formation contacts, our results also aid in constraining which areas should be drilled first to maximize  information gain.

For deep applications (e.g., depth to sedimentary basement or depth to Moho), our models require Bayesian incorporation of additional geophysical and geological data, such as active seismic (Johnson et al., 2013), passive seismic (Zhu and Kanamori, 2000) or drill hole geological observations (Beardsmore et al., 2016; McCalman et al., 2014). The incorporation of structural measurement at the surface and in drill core (e.g., faults, folds) could also aid in in informing the prior, particularly when seismic data is unavailable, to provide geologically-feasible models at the surface and shallow subsurface. Other geophysical surveys (e.g., magnetotellurics, radiometrics) could significantly improve the model certainty by identifying other variables in which geological units can have different rock properties.



The similarity of geophysical responses from different geological units in terranes that are broadly granitic (e.g., Halfway Gneiss and Durlacher Supersuite) has meant that the time to reach convergence is significantly greater than studies with units that display vastly different rock properties (Beardsmore et al., 2016; McCalman et al., 2014). The limiting factor is the ability

to explore very high-dimensional posteriors that result from a large-scale non-parametric model (i.e. the number of control points at a given resolution scales exponentially with area). Our modelled area is $13.5 \times 13.5$ km, which is useful for local-scale mineral exploration or detailed geological mapping, but may not be useful for reconnaissance-scale mineral exploration or terrane-scale geological modelling. Although our convergence times are not prohibitive for up-scaling the model to, for example $100 \times 100$ km, computational time becomes difficult for developing 3D geological models for significantly larger

areas (e.g., the entire Gascoyne Province). Incorporation of other data types (see above) may be part of the solution but these all rely on hand sample petrophysical measurements, which are not routinely collected, let alone often reported in the geosciences. To solve the paucity of petrophysical data compared to geophysical surveys, the MCMC sampler could be modified to a reversible jump scheme (Green, 1995; Sambridge, 2013), which is able to define the number of layers, sampling over rock categories to define a baseline prior irrespective of available rock property data.

An important limitation relates to the confidence of geological field observations. This study has simplified the probability distribution of each field observation to a single beta-binomial distribution, when different supporting data (age, geochemistry, sample descriptions) will provide different likelihoods. Probability distributions of samples that have, for example, age and geochemical data, should be significantly more confident than samples that only have hand sample descriptions. However, the

exact range of probabilities to ascribe to these samples still requires some user input. To make this process Bayesian and fully remove operator bias in assigning probabilities to field observations, an independent study will need to be conducted that purely assesses the likelihood of geological field observations, taking into account information like age data, geochemistry data and sample descriptions.

Another limitation of the current model is that only two geological units are modelled. In this study, the rationale for such a simplification is that the Halfway Gneiss and Durlacher Supersuite comprise >90% of the surface geological units (Fig. 2) but this will rarely be the case in other problems. Integration of volumetrically-minor geological units may be vitally important with respect to mineral exploration or unravelling tectonic histories (e.g., Li-bearing pegmatites; Kesler et al., 2012). A major impediment to effectively modelling these volumetrically-minor units is the line spacing for different geophysical surveys and

the potential paucity of geological field observations. In areas that are covered by shallow regolith or sedimentary cover, the problem of modelling volumetrically-minor units is exacerbated due to unavailability of geological surface measurements. Here, the only solution is to have drill hole geological information. Integration of drill hole geological units is already possible to build into the modelling process but moderately deep (>100 m) drill hole data is lacking for the modelled part of the Gascoyne Province.



## 6        Conclusions

Bayesian integration of geological field observations with geophysical survey data yield statistically-reliable and geologically-plausible 3D models at the surface and shallow subsurface (<4 km). Approximately 89% of the model area has >95% certainty.

Regions of <95% certainty are found exclusively within 1 km of mapped or inferred geological boundaries. The widths and positions of regions with <95% certainty are primarily a consequence of lack of geophysical, petrophysical or geological data. Our results indicate that these widths of these uncertain regions can be reduced by targeted geophysical surveys, petrophysical data collection and/or geological mapping. The integration of drill hole geological data and geophysical surveys with higher wavelengths (e.g., active seismic) are required to model deeper into the Earth's crust. Ultimately, the fusion of surface

geological observations with geophysical data yield robust 3D geological models with narrow uncertainty regions at the surface and shallow subsurface that will be especially valuable for mineral exploration and the development of 3D geological models under cover.

### Code and data availability

Aeromagnetic survey (Geological Survey of Western Australia, 1996) and gravity (Mathews and Jecks, 2010) data and

metadata is freely available from the Geophysical Archive Data Delivery System (www.geoscience.gov.au/geophysical-data-delivery). Petrophysical data are from Aitken et al. (2014). Geological field observations are available from WAROX, a database managed by the Geological Survey of Western Australia (GSWA). Geological field observation data may be directly requested from GSWA.

Model code is stored on GitHub (link to be confirmed)

**Author contribution**

HKHO, RC, CC, SR and RDM designed the project. RS and DK developed the experiments, model code and performed the simulations. HKHO prepared the manuscript with contributions from all co-authors.

### Acknowledgements

This research was funded by the Science and Industry Endowment Fund as part of The Distal Footprints of Giant Ore Systems: UNCOVER Australia Project (RP04-063) — Capricorn Distal Footprints.





## Competing interests

The authors declare that they have no conflict of interest.

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



**Figure Captions**

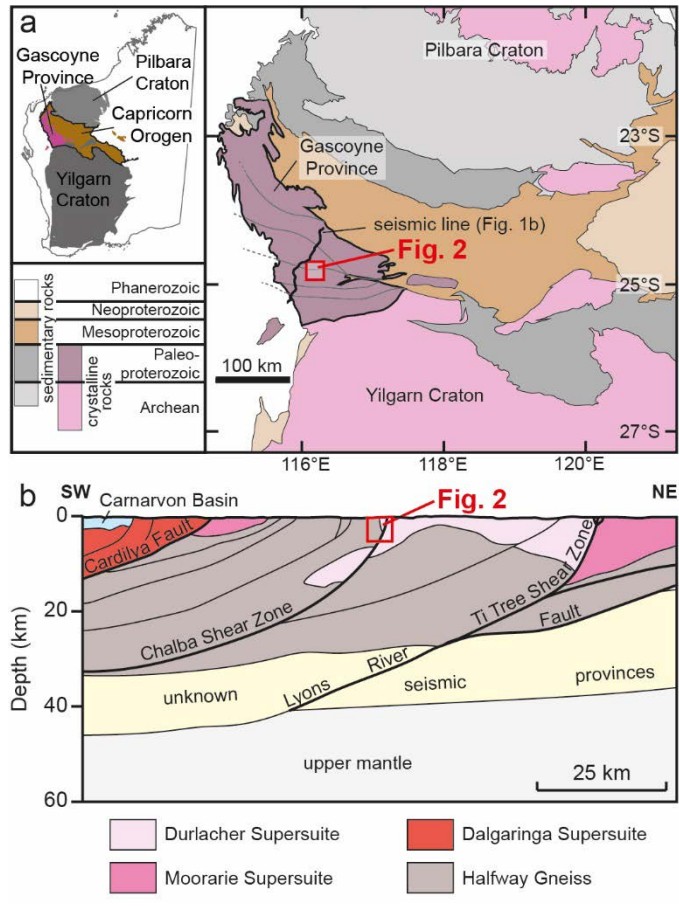

**Fig. 1: (a) Geological map of the West Australian Craton, modified from Sheppard et al. (2016), showing location of seismic section. (b) Interpretation of part of seismic line 10GA-CP2, after Johnson et al. (2013). The modelled region in this study is shown on both the map and seismic section (cf. Fig. 2).**





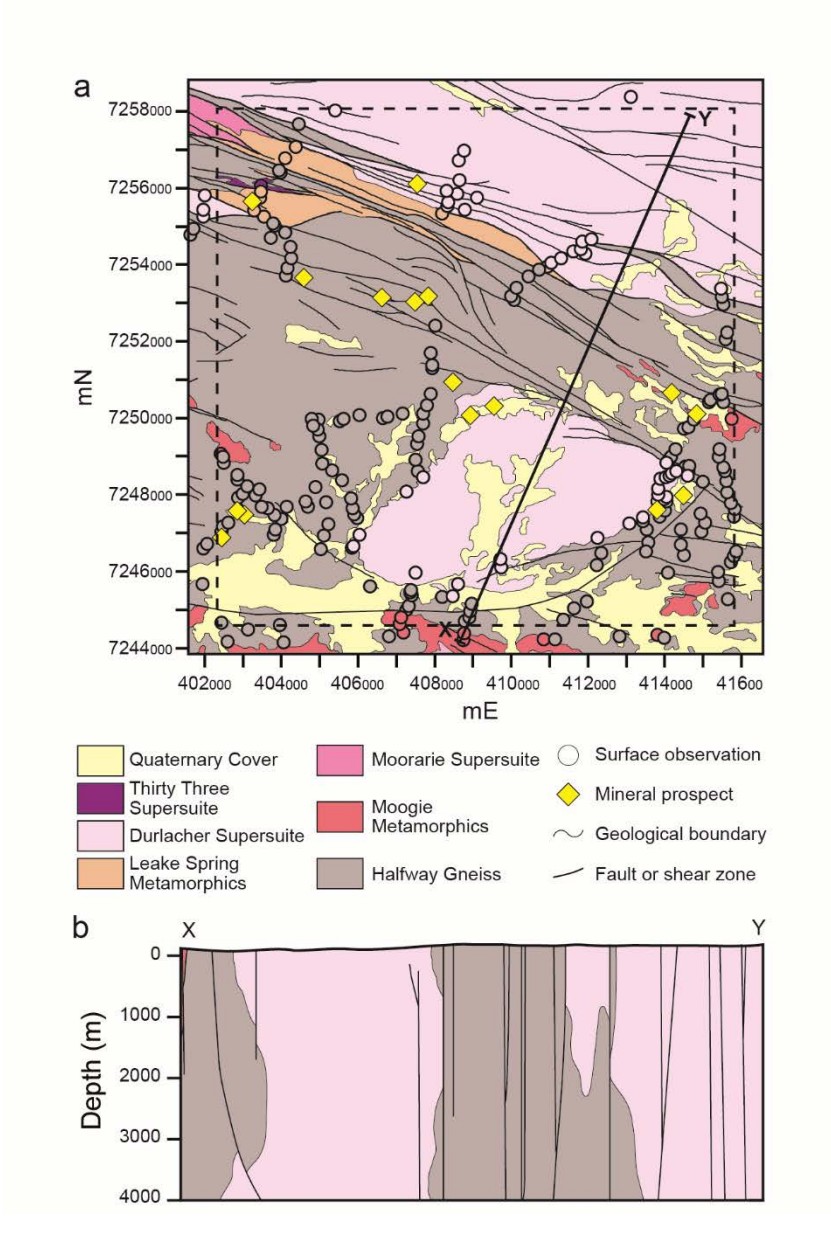

**Fig. 2: (a) Detailed geological map of a 15 × 15 km portion of the Gascoyne Province with the same centre as the modelled 13.5 × 13.5 km area (dashed area), showing geological units, structural discontinuities (faults/shear zones), geological surface observations and mineral prospects and deposits. (b) Cross-section through detailed geological map. Map and cross-section compiled using 1:100,000 geological maps from the Geological Survey of Western Australia.**



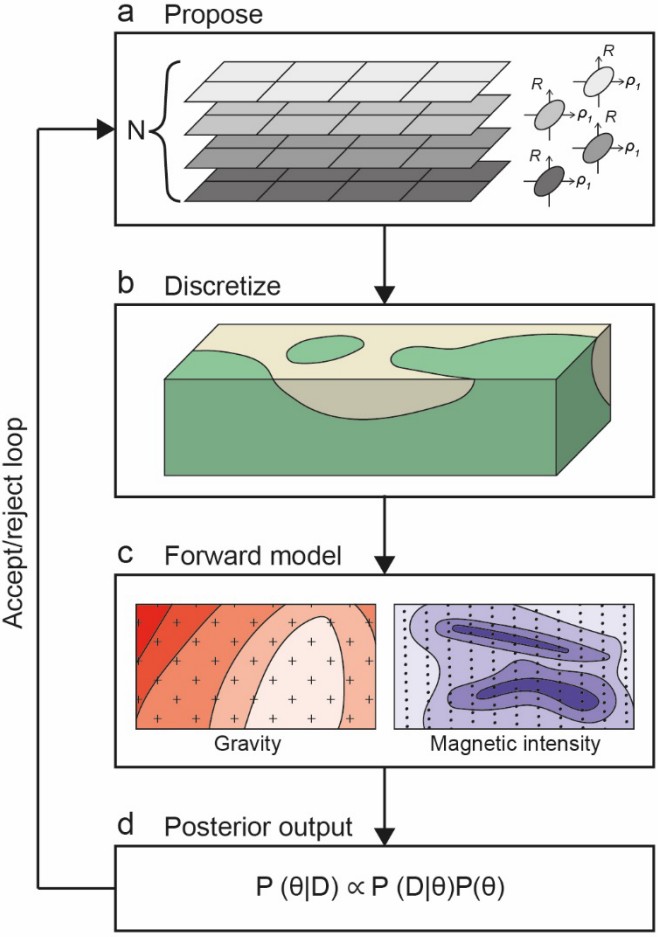

**Fig. 3: Obsidian workflow. (a) A set of world parameters, including geometry and rock properties for each rock formation included in the 3-D model, is proposed at random representing a small departure from the current parameter set. (b) These parameters are then rendered to form a discretized 3-D model, from which (c) forward models are calculated to make predictions for the expected readings in each sensor. (d) In the final step, the likelihood P(D|θ) of the observed dataset given the parameters is calculated, together with the prior probability P(θ) assigned to those parameters. The sample is then accepted or rejected according to the Metropolis-Hastings criterion, and the process begins again at (a). Over time the samples from this process will be distributed according to the posterior probability P(θ|D).**

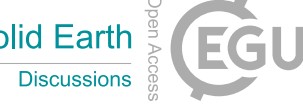



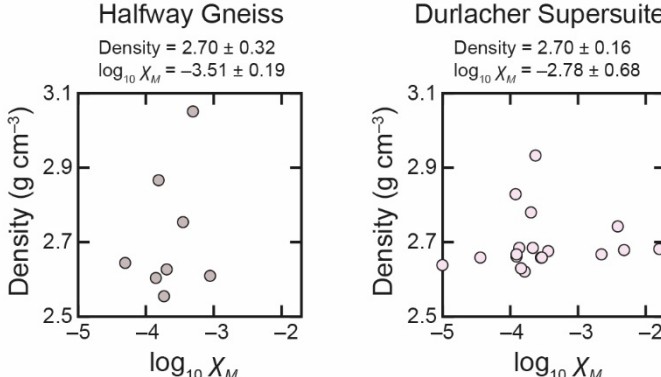

**Fig. 4: Measured density and magnetic susceptibility ($\log_{10} \chi_M$) for modelled geological units: (a) Halfway Gneiss, (b) Durlacher Supersuite. Mean density and susceptibility errors are quoted at 2σ uncertainty.**



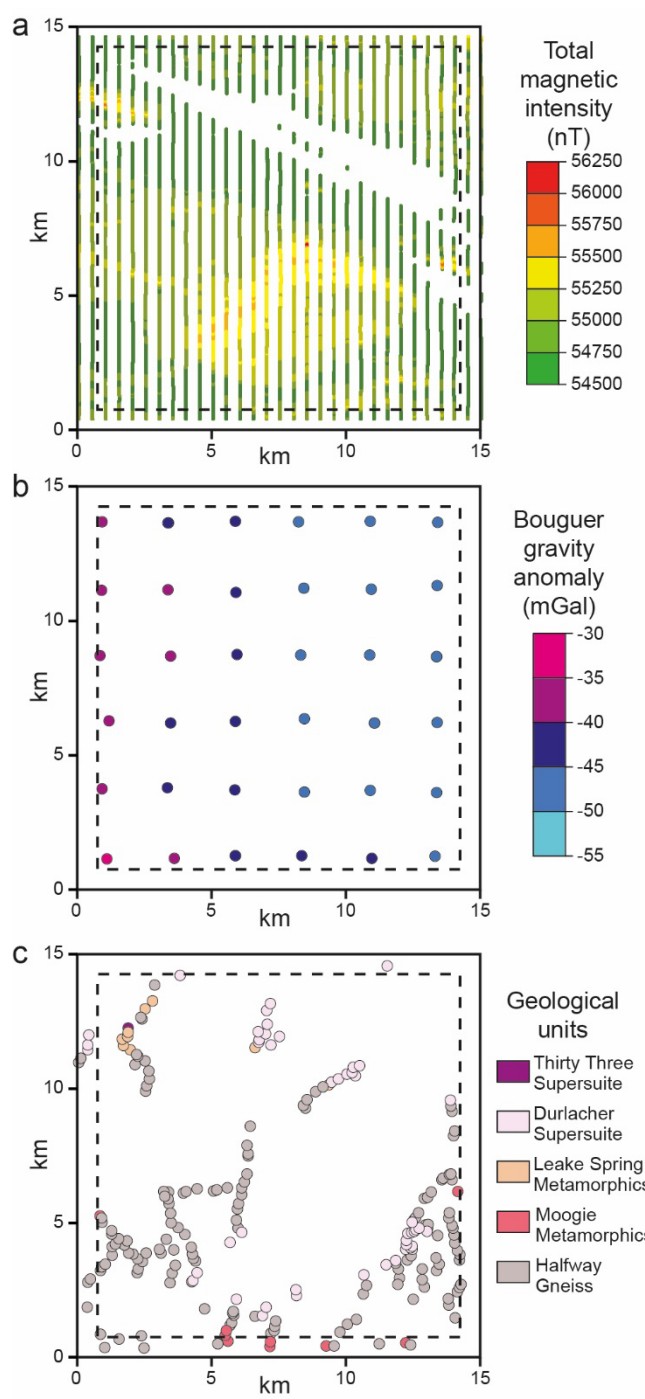

**Fig. 5: Measured geophysical survey data and geological field observations for 15 × 15 area, with dashed line representing modelled 13.5 × 13.5 km area. (a) Aeromagnetic data, showing locations of measured data on flight lines. (b) Gravity data, showing locations of ground-based measuring stations. (c) Geological field observations. Note the paucity of units other than the Halfway Gneiss and Durlacher Supersuite.**



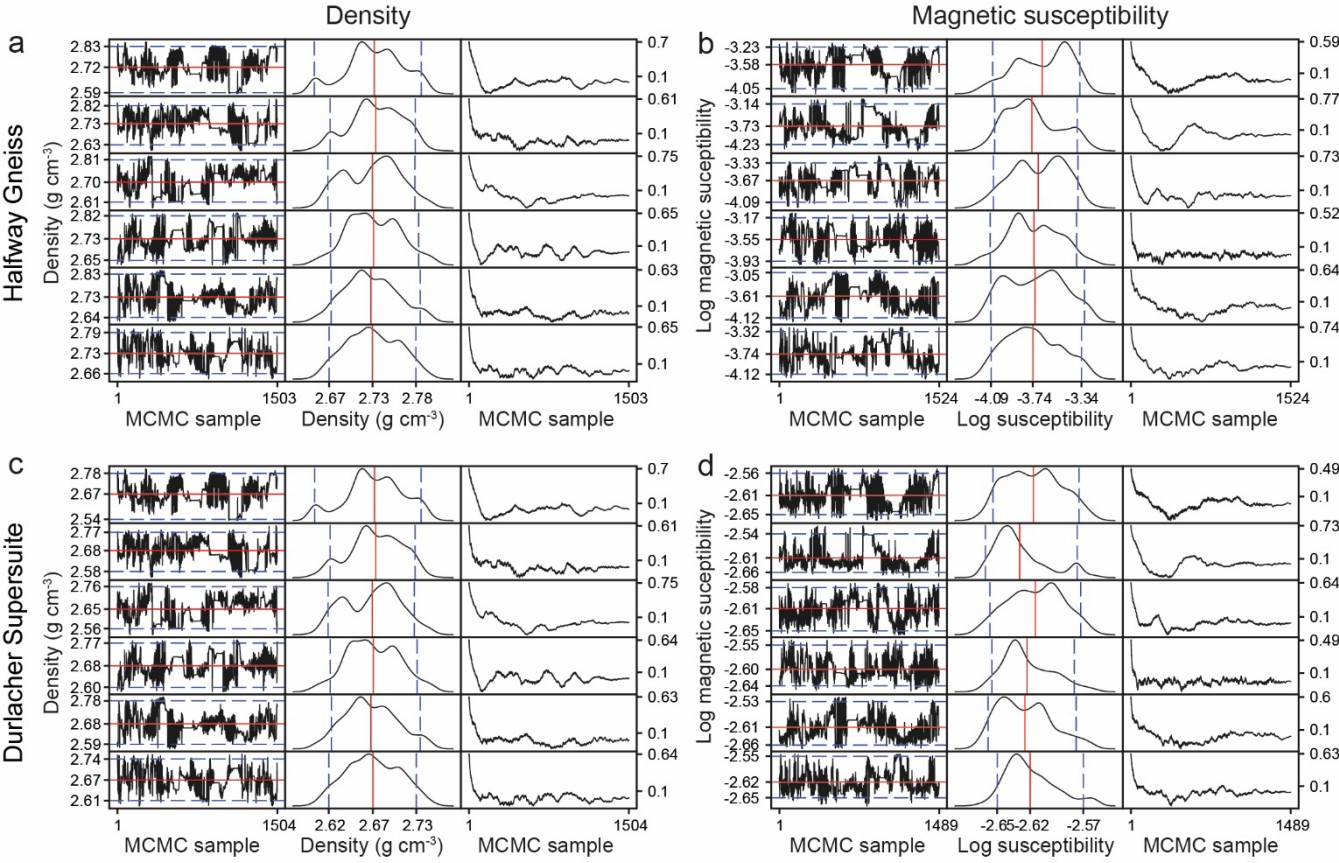

**Fig. 6: MCMC diagnostics for the modelled Halfway Gneiss (a,b) and Durlacher Supersuite (c,d), for density (a,c) and magnetic susceptibility (b,d). Column 1 in each panel shows six of the twelve lowest temperature chains. Column 2 in each panel shows the distribution of petrophysical properties per chain. The red line and blue lines in columns 1 and 2 are the mean and 2σ, respectively. Column 3 shows the autocorrelation time from the beginning of each chain to the end. Columns 1 and 3 MCMC iterations are thinned by 1000 (i.e., total number of samples is approximately 1.5 million per chain).**





**Fig. 7: Geweke scores for modelled densities and magnetic susceptibilities for the Halfway Gneiss and Durlacher Supersuite, shown for six chains.**





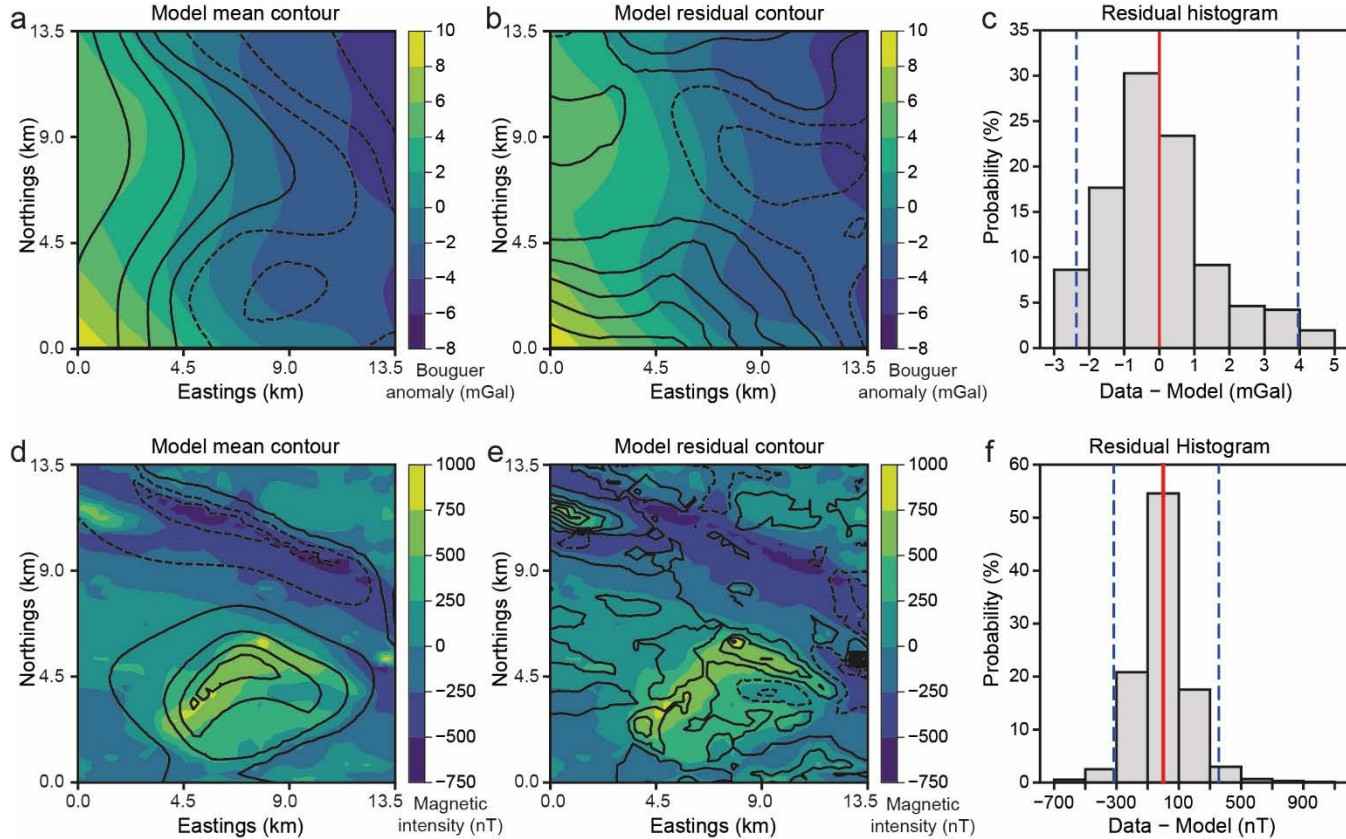

**Fig. 8: Modelled Bouguer anomaly and magnetic intensity.** Modelled mean contours of (a) Bouguer anomaly and (b) magnetic intensity compared to interpolated mean colored data. Modelled residual (i.e., data – model) contours for (c) Bouguer anomaly and (d) magnetic intensity compared to interpolated mean colored data. In a–d, contour lines are in 2 mGal and 250 nT increments for gravity and magnetic intensity, respectively, where solid lines ≥ 0 and dashed lines < 0. Histograms of residuals for (e) Bouguer anomaly and (f) magnetic intensity.





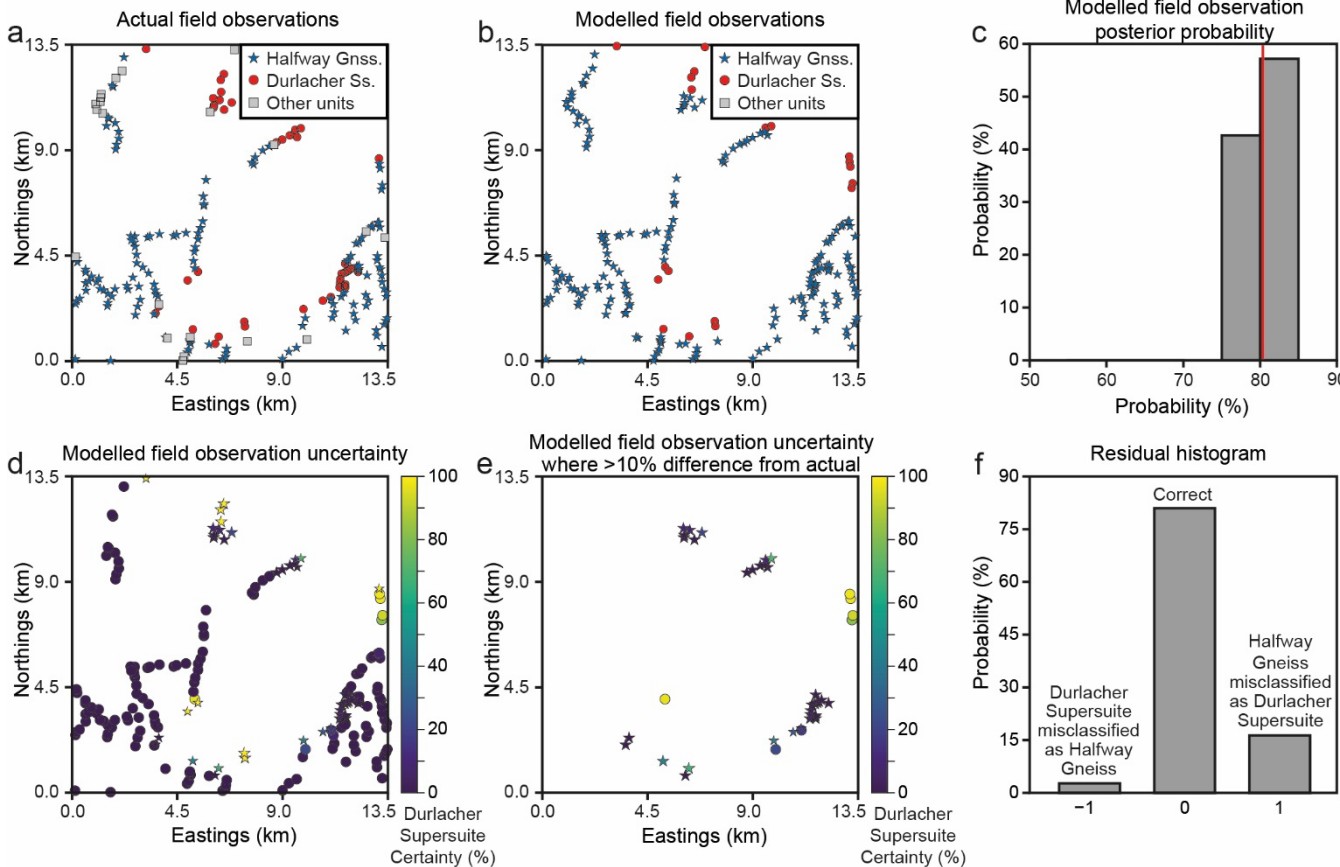

**Fig. 9: Actual vs. modelled field observations. (a) Actual field observations. (b) Modelled field observations, showing highest probability geological unit. (c) Modelled field observation posterior probability. (d) Modelled field observations with uncertainty. (e) Modelled field observations with uncertainty, where >10% different from actual field observations. (f) Residual histogram, where −1 is Durlacher Supersuite misclassified as Halfway Gneiss, 0 is correctly classified and 1 is Halfway Gneiss misclassified as Durlacher Supersuite.**





**Fig. 10: Voxelized posterior distributions of the 3D geological model compared to simplified GSWA maps and cross-sections. (a) Simplified geological map from Figure 2, showing interpreted boundaries for the Durlacher Supersuite, Halfway Gneiss and other formations (undifferentiated). (b) Model of the surface. (c) Interpreted geological cross-section through X–Y in (a). (d) Modelled cross-section through X–Y in (b).**



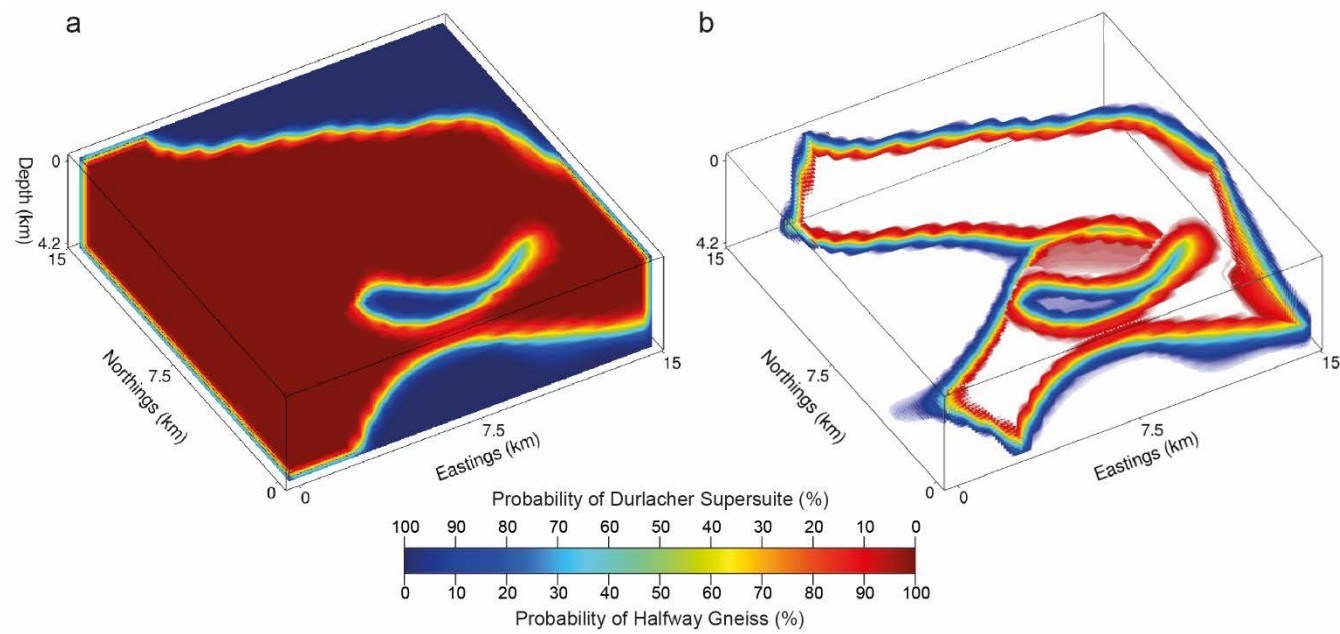

**Fig. 11: Voxelized posterior distributions in 3D, looking towards the northeast. (a) Probability of Halfway Gneiss and Durlacher Supersuite. (b) Same as (a) but only showing regions between 5 and 95% probability (i.e., the interface between Durlacher Supersuite.**

| Formation, petrophysical property | Mean | −2σ | +2σ | Autocorrelation time | Effective n | R̂ |
|---|---|---|---|---|---|---|
| Halfway Gneiss, Rock Density (mGal) | 2.72 | 0.13 | 0.11 | 11.61 | 129.1967 | 1.02 |
| Halfway Gneiss, Log Susceptibility | −3.65 | 0.53 | 0.57 | 14.26 | 105.2075 | 1.03 |
| Durlacher Supersuite, Rock Density (mGal) | 2.67 | 0.13 | 0.11 | 11.59 | 129.4099 | 1.02 |
| Durlacher Supersuite, Log Susceptibility | −2.61 | 0.05 | 0.09 | 13.32 | 112.6397 | 1.02 |

10   **Table 1: Rock property diagnostics for density and magnetic susceptibility for both the Halfway Gneiss and Durlacher Supersuite.**