# Peer review of "Bayesian geological and geophysical data fusion for the construction and uncertainty quantification of 3D geological models"

_Solid Earth, 2019_

## Referee Comment (RC1) · Anonymous Referee #1 · 12 Feb 2019

**1. General comments**

Olierook et al. present a study where they integrate information from geological maps and observations, petrophysical measurements and geophysical data. They focus on a small area in Western Australia and use interpretation from a nearby seismic survey to constrain their modelling. Their aim is to present an example of holistic inverse modelling where gravity and magnetic data are constrained using all the available information.

There is some novelty in their approach and the study they show can be published after the authors address a series of comments about specific issues in the text and more

general issues which, at the moment, is problematic. To date, Monte Carlo approaches for geological uncertainty have focussed on regional scale studies, while the work is clearly at a smaller scale. I think that this should be highlighted as one of the novelties of this paper.

I think that the formulation of the geophysical inversion problem should be described in more detail and that giving essential equations about geophysics and uncertainty assessment would improve the manuscript greatly. The only equation shown in the paper does not suffice to provide a good understanding of the basic mechanics of the methodology. Besides, after a quick manual derivation, I think that the right hand side of the equation provided might not be correct and may need revision. In any case, this derivation needs to be justified, by invoking an Eulerian integral of the first kind and the Beta function, using either the appropriate reference(s) and providing a succinct appendix.

Some references may be missing or are mis-cited. Several studies have been overlooked and have not been cited. This comment is relatively minor but addressing it would be important to show where the presented work stands in the literature. In my view, the introduction should emphasize the fact the idea of geologygeophysics integration is not new but that quantitative integration of both discipline is an area of research that has received more and more attention recently. Some references need to be added, but I will come back to this in my detailed comments of the document. Please check that all the papers you cite as 'in review' are still in review and have not been published.

The hypotheses and assumptions that the authors made need to be clarified. They neglect the presence of some geological units on the basis that their relative coverage in previous interpretation amount to only a few percent of the total. This is a simplification that need solid justification as it is sometimes the case that only a very small portion of a rock unit of interest is outcropping.
Moreover, the area of the authors' study is known to be prospective for several minerals, the deposits of which is often not born by geological units making up most of the geology of the area, and are covered by regolith and outcrop only at a few locations.

An aspect of geophysical joint inversion that needs to be considered is the relative impact of the two different geophysical datasets inverted for when they present largely different spatial coverage and sampling. This is not clearly mentioned in the manuscript and I expect that there would be an imbalance as the magnetic dataset seems to have about 100x more points than the gravity one (see Kamm et al. (2015), Sun and Li (2016)). How do you cope with the fact that in such case your joint inversion may be dominated by magnetic data? State it clearly. The Figures are not all very informative and some could be grouped. Fig. 1 and Fig. 2 both refer to the geology of the area and are referred to next to each other in the manuscript. It may be a good idea to merge them. Fig. 3 is not very informative. In Figure 5, I think that it would be good to have the line of cross-section X-Y shown.

The authors rely a lot on Scalzo et al., in review, which is a good complement to the manuscript. However, it is submitted to a different journal and is still in review. For this reason, I suggest that they reduce the dependency of their manuscript to Schalzo et al., in review, and explain succinctly key concepts they refer to Scalzo for explanation. This would make the paper more readable and easier to understand as all key elements would be readily available. The manuscript also has a number of sentences or pieces of sentences that are either an exact match or are very close to what can be read in Scalzo et al., in review. This is not an accusation of plagiarism but a mere observation. There are a few occurrences that I noticed when reading Scalzo et al., in review.

Below are the detailed comments I have.

2. Detailed comments and technical corrections

You mention the fact that you use a global optimization technique only in 2.2. Maybe state it in the intro.
P1 I14: "model results": results of the technique introduced here? Could be clearer.

P1 I16-18: "The boundaries between geological units are characterized by narrow regions with

robust geology-geophysics integration. I suggest to cite some of these as they strongly advocates for the kind study presented here.

P2 I31-33: "One useful addition to the current features of Obsidian would be the integration of geological and geophysical field observations made on the Earth's surface, which are vital for surface and near-surface applications (

unclear as Giraud et al derive constraints from a collection of geological models, therefore using all realizations from MC sampling of geological model space. But because their inversion is deterministic, they obtain a single geolophysical inverse model, which indeed represent a single scenario. I think that this is what you mean but it needs to be clarified.

P4 I21-30. "The Bayesian framework converts a deterministic model into a probabilistic one by using probability distributions to represent the free parameters rather than using optimal or single-point estimates." To make this clear and unambiguous, you should remind the principle behind Bayesian approaches, when you invoke Bayes' theorem. Solving a problem in a probabilistic way does not necessarily make it Bayesian. In this sentence you may want to stress the fact that you also use the prior distribution and sample the posterior, as it is a major difference with deterministic inversions. The utilisation of the priors is stated in the next sentence but I think that it could be made clearer overall. I would also not cite Oldenburg 2005 here, but perhaps one of A. Tarantola and others' publications which are seminal to many inversion approaches.

P5 I1: I strongly suggest that you add the mathematical foundation of your Bayesian inversion methodology. Just a few of the equations centre to your modelling approach would do and I think be informative to the reader. P5 I10: "convergence can be challenging" rephrase. Something like: "convergence can be difficult to reach"

P5 I13-24: Please add other cites to your citations of works by M. Sambridge and his team's. For instance, they may have brought lots of new ideas and methods to the field but the Metropolis-Hastings algorithm is not by them.

P5 I23: section 2.3. You need to give more information about Obsidian. It is too short and not sufficiently informative. The main reference you use (Schalzo et al in review) is still in review a basic summary of Obsdian should be self-contained in the paper.

P5 I31-32: This sentence is exactly the same as Scalzo et al., in review, beginning of section 2.4. Layers are not discrete if they have smooth boundaries. I suspect that
what you try to say is that the outline of a given layer is not angulous (i.e., that its representative functional would be differentiable)? If so this should be made clear.

P6 I2-4: "The layer boundaries are indexed in a strict order of increasing depth in the subsurface but are permitted to cross." The fact that they cross violates basic stratigraphic principles. You need to state explain briefly how you cope with that and how this is dealt with by your algorithm. This sentence is exactly the same as Scalzo et al., in review before they introduce equation 9.

P6 I10: "x and y correlation length" you haven't introduced what x and y are, except in Fig. 2. Just say that your RBF is anisotropic and I think that it's enough. The first sentence of this paragraph is very close to the one preceding equation 10 in Scalzo et al., in review.

P6 I13-15: please reformulate the last sentence. The parameter alpha and beta haven't been defined.

P6 I18: PTMCMC: this acronym has not been defined yet.

P6 I32: "and cross-cuts" you cannot rule out all uncertainty about the fact that it crosscuts, but you are making the (reasonable) assumption that it does. Please add this information.

P7 I1: "ordering of layers". Say "stratigraphy" or "stratigraphic pile".

P7 I6-9: typo in line 6. I don't think that a lithology with 3% of occurrence is insignificant. You need to provide more information as to why you do not account for rock units except Halfway Gneiss and Durlarcher Supersuite.

P7 I14-16: how do your values compare to litterature values, or work reported in the area or similar settings? If you have only approx. 100 samples to characterize several rock units through mean and standard deviation you can assume that your uncertainty on these parameters is quite high.

SED
P7 I24-32 and 1st paragraph in P8: I don't think that it is necessary to give detailed information about the processing of the field measurements. You can keep it, but I recommend to put it in Appendix instead of the full text.

P8 I15: consider replacing 'acquired' by 'obtained'.

P8 I21: I think that you should replace ">" by plain text words.

P9 I2-3: please explain briefly how the shape parameters were obtained.

P9 I9: please number the equations.

P9 I13-14: the calculation of this integral is not straightforward. If you want to leave it in the manuscript as it is please add a short appendix explaining how it is calculated, and at least provide the appropriate references.

P9 I17: please define what 'ID dataset' is.

P9 I25: the 'iGRW' acronym is not used in the rest of the text. Delete.

P10 paragraph 1: please add a figure to help the reader to understand how this works. The readership of Solid Earth might not be specialist in MCMC techniques.

P10 I 13: This information is relevant only if you provide information about the computing resources you used.

P10 I17: please add in the methodology the definition of the indicators you use to analyse your results.

P10 I22: I can make an educated guess about what sigma means here, but it needs to be clearly stated.

P11 I13: the reference to Geweke score should come earlier in the text. The ref given might not be the best. I would cite the following instead: Geweke, Evaluating the accuracy of sampling based approaches to the calculation of posterior moments, 1992, Bayesian Statistics 4, pp. 169-193
P11 section 4.2. Although it's not perfect as an indicator, I'd also give the root-meansquare error. As said above I do not provide review comments on the results section. From here I go straight to section 5.2.

P14 I20-21: "So, if higher resolution geophysical surveys and/or geological field observations are acquired, the model can then become more precise". This is not necessary as it is obvious.

P14 I24-25: "Where such regions are under cover and drilling is required to establish formation contacts, our results also aid in constrainingwhich areas should be drilled first to maximize information gain". This is true as a first approximation but not always valid. You can imagine that adding more information in a certain part of the model may improve greatly a portion of the model that is equally uncertain because it is linked to that first structure in a structural or topologic sense.

3. References

Bijani, R., P. G. Lelièvre, C. F. Ponte-Neto, and C. G. Farquharson, 2017, Physicalproperty-, lithology- and surface-geometry-based joint inversion using Pareto Multi-Objective Global Optimization: Geophysical Journal International, 209, 730–748.

Fullagar, P.  $\sim$ K., G.  $\sim$ a. Pears, and B. McMonnies, 2008, Constrained inversion of geologic surfaces - pushing the boundaries: The Leading Edge, 27, 98–105.

Guillen, A., P. Calcagno, G. Courrioux, A. Joly, and P. Ledru, 2008, Geological modelling from field data and geological knowledge. Part II. Modelling validation using gravity and magnetic data inversion: Physics of the Earth and Planetary Interiors, 171, 158–169.

Jessell, M., E. Pakyuz-charrier, M. Lindsay, J. Giraud, and E. de Kemp, 2018, Assessing and Mitigating Uncertainty in Three-Dimensional Geologic Models in Contrasting Geologic Scenarios: , 63–74.

Jessell, M., L. Aillères, E. De Kemp, M. Lindsay, F. Wellmann, M. Hillier, G. Laurent, T.
Carmichael, and R. Martin, 2014, Next Generation Three-Dimensional Geologic Modeling and Inversion: SEG Special Publication 18, Chapter 13, 261–272.

Jessell, M. W., L. Ailleres, and E. A. de Kemp, 2010, Towards an integrated inversion of geoscientific data: What price of geology? Tectonophysics, 490, 294–306.

Kamm, J., I. A. Lundin, M. Bastani, M. Sadeghi, and L. B. Pedersen, 2015, Joint inversion of gravity, magnetic, and petrophysical data — A case study from a gabbro intrusion in Boden, Sweden: GEOPHYSICS, 80, B131–B152.

de la Varga, M., A. Schaaf, and F. Wellmann, 2018, GemPy 1.0: open-source stochastic geological modeling and inversion: Geoscientific Model Development Discussions, 1–50.

Pakyuz-Charrier, E., M. Lindsay, V. Ogarko, J. Giraud, and M. Jessell, 2018a, Monte Carlo simulation for uncertainty estimation on structural data in implicit 3-D geological modeling, a guide for disturbance distribution selection and parameterization: Solid Earth, 9, 385–402.

Pakyuz-Charrier, E., J. Giraud, V. Ogarko, M. Lindsay, and M. Jessell, 2018b, Drillhole uncertainty propagation for three-dimensional geological modeling using Monte Carlo: Tectonophysics.

Scholl, C., J. Neumann, M. D. Watts, and S. Hallinan, 2016, Geologically constrained 2D and 3D airborne EM inversion through cross-gradient regularization and multi-grid efficiency: ASEG-PESA-AIG 2016, 1–6.

Sun, J., and Y. Li, 2016, Joint-clustering inversion of gravity and magnetic data applied to the imaging of a gabbro intrusion: SEG Technical Program Expanded Abstracts 2016, 2175–2179.

Wellmann, J. F., M. de la Varga, R. E. Murdie, K. Gessner, and M. Jessell, 2017, Uncertainty estimation for a geological model of the Sandstone greenstone belt, Western Australia – insights from integrated geological and geophysical inversion in a Bayesian Interactive comment

inference framework: Geological Society, London, Special Publications, SP453.12.

Zheglova, P., P. G. Lelièvre, and C. G. Farquharson, 2018, Multiple level-set joint inversion of traveltime and gravity data with application to ore delineationŕ: A synthetic study: , 83.

SED

---

## Referee Comment (RC2) · Anonymous Referee #2 · 13 Feb 2019

I thank the editor for inviting me to review this paper. Unfortunately, I find this paper quite underwhelming. The paper describes very little that has not already been shown in previous works, nor do the results convincingly reveal new understanding from the region. It is thus difficult to understand what contribution this study makes either to probabilistic methods in geophysics, or geological understanding of the Gascoyne Province. This is compounded by the authors' inadequate review of existing work and failure to place theirs in context with the discipline. The authors make statements about the "importance" or their results with no justification, and how that their method "is the only technique that provides a range of solutions" which is false. To reiterate, the authors needs to spend more time reviewing the existing literature.

[Figure]

One positive is that the manuscript is well-written and structured. I suspect the authors will be able to remedy many of the deficiencies listed here and produce an adequate revision.

I list the major issues directly below, followed by relatively minor comments.

Major comments.

Valid criticism is made of existing work in the Introduction (P2, L9-14), and refers to how "these approaches still require a significant degree of human decision making into how to fuse disparate geoscientific datasets." And "these approaches still largely elide the question of how the joint distribution of such parameters is meant to be derived." This infers the manuscript will then address these important issues, which it barely does. These statements are then followed by another which claims the presented method "will fuse all available constraints in a probabilistically rigorous fashion." The method doesn't fuse all available constraints (see discussion, where this is admitted), in fact it only uses a small subset of available data. One major omission is structural and drillhole data, which is used or can be used in all the methods described in the papers cited in this paragraph. These claims are at best poorly made, and at worse false. Pakyuz-Charrier and Giraud both address the issues of how joint distribution of parameters (geophysics, drill holes, petrophysics) are made. A far better justification of these statements needs to be made in order to emphasise the contribution of this paper to advancing this important area of research.

Comparing models results with maps. If the maps were made using interpretations from geophysics, then the model, which is based on geophysics, matching the map is not surprising, and expected, and thus not an adequate validation exercise. Please better justify the validation method.

How is the geological model built? Figure 2 implies five units are modelled, and then it's revealed deep into the discussion that only two were modelled. Differentiating between two geological units is not that exciting, not useful, especially at the scale of the study,

so the authors need to show better justification as to how this method is novel, and worthy of publication.

Two important papers that are not referred to and are very relevant to this work are:

Wellmann, J. F., M. de la Varga, R. E. Murdie, K. Gessner and M. Jessell (2018). "Uncertainty estimation for a geological model of the Sandstone greenstone belt, Western Australia – insights from integrated geological and geophysical inversion in a Bayesian inference framework." Geological Society, London, Special Publications 453(1): 41-56.

Guillen, A., P. Calcagno, G. Courrioux, A. Joly and P. Ledru (2008). "Geological modelling from field data and geological knowledge: Part II. Modelling validation using gravity and magnetic data inversion." Physics of the Earth and Planetary Interiors 171(1-4): 158-169.

Both these works describe methods similar to that being described here and deserve a good review in this paper. In particular Wellman et al. 2017 presents a Bayesian framework for geophysics that authors would benefit from during their review.

No figure shows any 3D model, either the initial, or geophysically constrained geological version, nor the inverted geophysical volume. This is a critical thing to show to the readers of Solid Earth, most of whom are geoscientists. How can we appreciate your endeavours without seeing the results, especially when "3D geological models" is in the title?

Downsampling of "geological" (really lithostratigraphic) observations. You have detailed "petrographic, geochemical and geochronological knowledge obtained on a subset of WAROX data" which would surely give far higher lithological resolution that the five bulk units that make your model (legend of fig. 5). How did you downsample these observations into the five major groups? As you hint, there is significant uncertainty, not just in correctly identifying the correct rock unit (though with the data you have this source of error should be reduced, but not irreducible). This alertoric uncertainty is inadequately

addressed in P9, L19. How did you determine the error in these observations? How was it translated into a Beta distribution? There is also the loss of information from the process of downsampling – i.e. epistemic uncertainty (which is reducible). You refer to section 3.4 in this matter, but section 3.4 barely describes this in a geological context. Other issues with section 3.4 exist. . . next paragraph.

Section 3.4 needs significant work. It is quite disjointed from the previous section. For example, how are the survey forward models determined? Why alpha = 1 and Beta = 2? There is no effort to make the text link with the previous sections, explain the importance of a Beta distribution to a Bayesian framework, nor appropriate translation of known uncertainties within a geological context or even related to widely understood sources of uncertainty in geoscientific data. In its current form this section is incomprehensible.

The results are not presented well. Section 4.2 Residuals from forward models: Statistics are presented without any indication as to whether they are acceptable (e.g. "Aeromagnetic residuals display an approximately Gaussian distribution of 0 +358 −31725 nT ($2\sigma$, 21% of the total magnetic range" – so what?) or even higher or lower than expected.

Section 4.3 Probability density of layer locations. Text associated with figure 9 states that rock observations near the contact between the Halfway Gneiss and Durlacher Supersuite are misclassified. None of this is very surprising given it's a contact which any geologist knows are hard to define. But the relevance of this finding is difficult to discern given the method for building the model isn't described anywhere, the cell sizes of the model are not given (see comments below), nor how the contact was defined in the first place. All it infers (given no other information) is that Obsidian doesn't manage to determine the geometry of this contact well. This may not be true, but none of the other results presented show that Obsidian has done a good job in this regard. This is not helped with the lack of description for geological model construction.

You state that results show the Durlacher Supersuite to be in two "domains". This isn't surprising given it is a Supersuite, and by definition made up of multiple suites, which can be defined as domains. This interpretation is also not supported by any geological data, nor is the importance of this made apparent during the introduction, discussion or conclusion. The results are described as being far more successful than they really are. Page 14, line 13: "Highly similar" – not really. There are a quite few differences, plus you have only shown the probabilities of two units, when the 3d model was built using 5 units (maybe? Again, describe how the model was built). What about the other three units? The assessment of this method is thus inadequate. As such much of the discussion unconvincing, especially when two select slices of the probabilistic model are shown.

You admit that structural data is not used in the discussion (P14, L29). This needs to be stated clearly in the method (where a description of model construction is required – see previous comments) and makes earlier comments criticizing previous work disingenuous (see major comments). It is self-evident that structural data is very useful for geological models. The use of structural data is shown in other methods that have been around for almost a decade (see uncertainty work by Wellmann, de la Varga, Bond, Lark, Lindsay, Jessell), or general modelling (see Calcagno paper). Why can you not do this? The same can be said for drillhole data, which other methods also use. The main problem is that both structural and drill hole data from the area is publically available, but not used. So it appears that Obsidian, or the described method cannot use these data presently, otherwise they would have done so. Other methods (as cited earlier) can use both structural, seismic and drillhole. How do the authors then justify this method as novel, or one people should adopt given is has severe limitations to inputs? Simply being Bayesian is not enough, especially as Bayesian methods are well suited to integration of different data types. Other Bayesian techniques have been proposed (de la Varga, Wellmann). This aspect needs a much fuller justification and discussion.

Minor Comments

Page 3, line 7: define "data-rich". Rich in diversity or coverage, both? Quantify this richness.

Page 4, line 11: technically measurements of gravitational acceleration and magnetic field strength

Page 4, line 16 – Be clear about the shortcomings of other work. Giraud et al. in review does acknowledge alternative geophysical, petrophysical and geological scenarios. Are you referring to alternative forms of parameterisation for regularization?

Page 4, line 27: it is unclear what you mean by "geophysical processes". Are you talking about how well the models represent the geology?

Page 6, line 18: "PTMCMC" define your acronyms before using them.

Subheading 3.1. "World" is an expansive term that infers all parameters, data, models, inferences, assumptions are under consideration in the following paragraph, which isn't true. "3D geological model parameterization" is more specific and less confusing. The same applies to all references of "world" models. This is important as you use and describe more than one model through the manuscript, including statistical, geophysical and conceptual are present as well.

Page 6 line 30: should be "magnetic susceptibility and density data"

Section 3.2: You need to show where these petrophysical data were acquired on a map (Fig. 2?). Presumably the petrophysical data locations will be different to the "surface observations" shown in Fig. 2. Given the caption describes them as geological surface observations

Page 7, line 21. Please describe what Bayesian "fusion" is, or just say the data were input to the Obsidian framework.

Page 7, line 31: explain the source of these "correlations", what they are correlated

with, and why this could be a problem. You do this later with the gravity data (P8,L10), so move that explanation here. But you still need to better explain the source of the biases and how they produce incorrect results in context of the Bayesian methods you describe earlier.

Page 9, Line 23 PT-MCMC or PTMCMC (as P6,L18)

Page 11, Line 18 Discussion of large Gweke scores

Page 11, Line 21 Figure 8 needs to show the measured interpolated image with the forward models for easy comparison, rather than forcing the reader to switch between figures on different pages. The reader is also referred to figure 2 when describing magnetic lineaments, but figure 2 is a geological map. Are the authors assuming the NW strike of the geology will also produce NW striking magnetic lineaments? This is a reasonable interpretation, but the authors need to first make that interpretation for the statement in line Page 11, Line 22.

Page 11, Line 25 – okay, but so what?

Page 11, Figure 8 caption. Labelling of figure parts appears to be incorrect. b) shows units in mGal, so not magnetic intensity, c) shows a histogram, not the modelled contours

Page 12, Line 14. Distances have no meaning without telling us what the model cell sizes are first. How many cells does 300-1000m represent?

Page 12, Line 14. "The ellipsoidal Durlacher Supersuite inlier is heterogeneously constrained." What does this mean and why is it relevant?

Page 13, Line 3 "a function of a long-wavelength (i.e. deep) gravity response" Careful here. A long wavelength is not always deep. It can be laterally extensive but shallow.

Page 13, Line 6. I would think more petrophysical data from "other geological units" (see previous sentence) would be more useful to define the lithostratigraphic diversity

than marginally tightening the standard deviation of the Halfway Gneiss and Durlacher SS units.

Page 13, Line 11-20 Annotate the figures with the various features being described here (Chalba SZ, Durlacher SS 'spur' and 'sliver' etc.)

Page 13, line 19-20. Tells me the initial 3D model is wrong.

Page 13, line 31. "Importantly, our method is the only technique that provides a range of solutions and quantitatively accounts for all the input assumptions" This grandiose statement needs far more justification. I actually think this should be removed entirely, given the technique is poorly described in the first place.

Page 14, line 1-9. Figure 10d? Over-interpreted results. "Definitively separated" Plus given the sections only extend to 4km, how can you be sure the Durlacher remains separated beyond that? Figure 10 d looks like the Halfway Gneiss only extends as far as 4km depth (which is also probably a function of the model volume parameters? Also needs discussing). You then state correctly in later lines (lines 4-5) that "it was difficult to know whether this spur of Halfway Gneiss between the two Durlacher Supersuite domains continued at depth or was truncated in the near subsurface". Hardly definitive! "This important contribution shows that small geological volumes on the scale of a few km can be resolved accurately and will be important when this modelling output is up-scaled to larger regions." Small volumes can be detected given appropriate geophysical data resolution and corresponding model parameters. Small volumes have also been detected by many other methods which I suggest you spend some time reviewing (Li and Oldenburg papers, Peter Fullagar, Guillen, etc etc) so you realise this is not a world first. Upscaling models to larger regions is also commonplace. If you are to make this kind of statement, please explain how upscaling should be done.

Page 14, line 29. How are drill holes going to help Bayesian methods >4km when they rarely extend beyond 400m?

Page 15, Line 22. Interesting concept, and I agree should be done, but expand on how this would be achieved?

---

## Referee Comment (RC3) · Gautier Laurent (Referee) · 8 Mar 2019

1. Does the paper address relevant scientific questions within the scope of SE? yes

2. Does the paper present novel concepts, ideas, tools, or data? yes

3. Are substantial conclusions reached? yes

4. Are the scientific methods and assumptions valid and clearly outlined? yes

5. Are the results sufficient to support the interpretations and conclusions? yes

6. Is the description of experiments and calculations sufficiently complete and precise to allow their reproduction by fellow scientists (traceability of results)? yes

7. Do the authors give proper credit to related work and clearly indicate their own new/original contribution? yes

8. Does the title clearly reflect the contents of the paper? yes

9. Does the abstract provide a concise and complete summary? yes

10. Is the overall presentation well structured and clear? yes

11. Is the language fluent and precise? yes

12. Are mathematical formulae, symbols, abbreviations, and units correctly defined and used? Mainly but some explanation seems to be missing and some badly located (cf annotated manuscript)

13. Should any parts of the paper (text, formulae, figures, tables) be clarified, reduced, combined, or eliminated? In places, some clarifications cf. comments below and in the attached document.

14. Are the number and quality of references appropriate? yes

15. Is the amount and quality of supplementary material appropriate? yes

**1 General comments**

The manuscript entitled "Bayesian geological and geophysical data fusion for the construction and uncertainty quantification of 3D geological models" presents an interesting approach for integrating geophysical and geological observations

into a probabilistic modelling approach of subsurface. Joining the gap between geological and geophysical inversion of subsurface has been a long-standing problematics and this study takes one step toward this objective. Being from the geological side of this problematic my main observations are that the geological description of the subsurface that is used in this study remains relatively simple (probably for sake of parsimony, as required by inversion problems). I would not place this remark as a criticism for this study but rather as an acknowledgment that further research is needed for improving the parameterisation of geological models.

This study has the merit of showing the interest of coupling geophysical and geological inversion. Therefore, I find it valuable for the community and I would recommend accepting it for publication provided some minor revisions.

**2 Specific comments:**

The parameterisation consist of a Gaussian process with an RBF interpolation of depth map of the geological contacts. You should better discuss the limitations of this approach on the geometry of the contacts. At least it cannot reproduce multi-z structured, but it would also have limitations for vertical or sub vertical contacts. As you mentioned in the introduction Obsidian was designed for basins, ie. With layers being roughly horizontal. Isn't it limiting the sampling and more generally the applicability of the method to other geological regions? The description of this geological parameterisation was apparently supposed to be supported by fig. 3 but several reference to this figure in page 6 line 1 and 10 are apparently not pointing at the right thing. The user-defined control points of the Gaussian process are not show as expected. In addition, it would be interesting to show the prior distribution for the parameters describing the depth of the geological interface, which has apparently been omitted.

You chose to ignore some lithologies based on their smaller cover of the surface. What is the mag sus and density of these formations? They are ignored (because barely seen on surface) but are they going to affect the magnetic and gravity field?

I think you should clarify the way you introduce and discuss you probabilistic approach of the lithological observations. Unless there are arguments for taking particular care with these observations, they seem to me to be rather hard constraints as compared to gravity and magnetic responses. Of course, observations could be misinterpreted, but unless the two discussed units are very similar, you would not need chemical analysis or dating to assign them to one or the other group. On the other hand, gravity and magnetic field are by nature ambiguous.

Why are the high probability areas that outcrop in the middle of the modelled region so different between fig 9b and 10a? On 9b it looks roughly circular whereas it has a crescent shape on 10a.

Please refer to the annotated manuscript for more detailed comments and corrections.

Please also note the supplement to this comment:
https://www.solid-earth-discuss.net/se-2019-4/se-2019-4-RC3-supplement.pdf

[Figure]

**Supplement:**

**Bayesian geological and geophysical data fusion for the construction and uncertainty quantification of 3D geological models**

Hugo K. H. Olierook1, Richard Scalzo2, David Kohn3, Rohitash Chandra2,4, Ehsan Farahbakhsh2,4, Gregory Houseman3, Chris Clark1, Steven M. Reddy1, R. Dietmar Müller4

[revised manuscript text omitted]

---

## Author Comment (AC1) · 3 May 2019

Reviewers' comments:

Reviewer #1: 1. General comments Olierook et al. present a study where they integrate information from geological maps and observations, petrophysical measurements and geophysical data. They focus on a small area in Western Australia and use interpretation from a nearby seismic survey to constrain their modelling. Their aim is to present an example of holistic inverse modelling where gravity and magnetic data are constrained using all the available information.

[Figure]

We thank the reviewer for their rapid turnaround time and detailed comments. Please find attached a zip file containing comments to all reviewers, a manuscript with track changes and a clean manuscript with all changes incorporated.

There is some novelty in their approach and the study they show can be published after the authors address a series of comments about specific issues in the text and more general issues which, at the moment,is problematic. To date, Monte Carlo approaches for geological uncertainty have focussed on regional scale studies, while the work is clearly at a smaller scale. I think that this should be highlighted as one of the novelties of this paper.

AGREE. We have now highlighted this in the abstract, introduction and discussion.

I think that the formulation of the geophysical inversion problem should be described in more detail and that giving essential equations about geophysics and uncertainty assessment would improve the manuscript greatly.

AGREE. We have added many of essential equations that were previously only present in Scalzo et al., in review. We also clarify the covariance function for boundaries and the prism approximation used in calculating potential fields (citing Li & Oldenburg).

The only equation shown in the paper does not suffice to provide a good understanding of the basic mechanics of the methodology. Besides, after a quick manual derivation, I think that the right hand side of the equation provided might not be correct and may need revision. In any case, this derivation needs to be justified, by invoking an Eulerian integral of the first kind and the Beta function, using either the appropriate reference(s) and providing a succinct appendix.

AGREE. We appreciate the close attention to detail. The form of the beta-binomial likelihood shown in the earlier version was indeed incorrect. We have verified that the correct version was used in our code; the error is one of transcription and not the underlying derivation. We now provide a more detailed form of that derivation in an

appendix, as well as the derivation for the Student's-t likelihood used for the gravity anomaly and magnetic intensity.

Some references may be missing or are miscited. Several studies have been overlooked and have not been cited. This comment is relatively minor but addressing it would be important to show where the presented work stands in the literature. In my view, the introduction should emphasize the fact the idea of geology-geophysics integration is not new but that quantitative integration of both discipline is an area of research that has received more and more attention recently. Some references need to be added, but I will come back to this in my detailed comments of the document. Please check that all the papers you cite as 'in review' are still in review and have not been published.

AGREE. We have added several more references throughout the manuscript, as suggested by reviewer 1 and 2. Only Scalzo et al. was still in review with GMD and, unfortunately, is still in review.

The hypotheses and assumptions that the authors made need to be clarified. They neglect the presence of some geological units on the basis that their relative coverage in previous interpretation amount to only a few percent of the total. This is a simplification that need solid justification as it is sometimes the case that only a very small portion of a rock unit of interest is outcropping. Moreover, the area of the authors' study is known to be prospective for several minerals, the deposits of which is often not born by geological units making up most of the geology of the area, and are covered by regolith and outcrop only at a few locations.

PARTLY AGREE. The other lithologies are less significant volumetrically, appearing primarily near the surface, and are also present areas smaller than our model can resolve. However, we agree that this was not well emphasized in the text. We have focused in this work on mapping the boundary at depth between the two major units. More detailed future work will undoubtedly move towards sampling of more detailed

models that can resolve finer features.

An aspect of geophysical joint inversion that needs to be considered is the relative impact of the two different geophysical datasets inverted for when they present largely different spatial coverage and sampling. This is not clearly mentioned in the manuscript and I expect that there would be an imbalance as the magnetic dataset seems to have about 100x more points than the gravity one (see Kamm et al. (2015), Sun and Li (2016)). How do you cope with the fact that in such case your joint inversion may be dominated by magnetic data? State it clearly.

We would consider the imbalance in the sizes of training sets to be a genuine asymmetry in available information from the two sensors. However, the reviewer has a point in that the magnetic data is available on a much finer spatial scale than our actual model parametrization can possibly resolve. This produces correlated residuals from our baseline model that result from model inadequacy instead of from random variation of measured sensor values from an underlying forward model. The effect is reduced by our Students-t likelihood relative to the usual Gaussian likelihood, i.e. by our relatively vague prior about the expected noise variance in each sensor.

The Figures are not all very informative and some could be grouped. Fig. 1 and Fig. 2 both refer to the geology of the area and are referred to next to each other in the manuscript. It may be a good idea to merge them.

DISAGREE. Figure 1 and 2 are difficult to merge into one because they use different colour schemes and have different aspect ratios of maps and cross-sections. We have attempted combining Figs. 1 & 2 previously but found it added more confusion when merged then when separated.

Fig. 3 is not very informative.

AGREE. We have removed Figure 3.

In Figure 5, I think that it would be good to have the line of cross-section X-Y shown.

AGREE. The cross-section line in Figure 5 has been added to each of the panels, and the caption updated.

The authors rely a lot on Scalzo et al., in review, which is a good complement to the manuscript. However, it is submitted to a different journal and is still in review. For this reason, I suggest that they reduce the dependency of their manuscript to Schalzo et al., in review, and explain succinctly key concepts they refer to Scalzo for explanation. This would make the paper more readable and easier to understand as all key elements would be readily available. The manuscript also has a number of sentences or pieces of sentences that are either an exact match or are very close to what can be read in Scalzo et al., in review. This is not an accusation of plagiarism but a mere observation. There are a few occurrences that I noticed when reading Scalzo et al., in review.

The recurring phrases are oversights on our part. We originally had one longer paper which was split at an early stage of editing into "methods" and "applications" papers, and the repeated text elements are left over from this stage. Scalzo et al. focuses largely on sampling from the posterior distribution under different priors, while this work focuses on a specific application and on introduction of the beta-binomial likelihood for lithostratigraphic measurements. We agree that each paper should be able to stand on its own and that no text should be re-used. We have taken steps throughout to reduce the interdependence of the two manuscripts while preserving complementarity.

Below are the detailed comments I have.

2. Detailed comments and technical corrections You mention the fact that you use a global optimization technique only in 2.2. Maybe state it in the intro.

AGREE. We have added a mention to Obsidian's parallel-tempered MCMC scheme to paragraph 3 in the introduction. We re-emphasize there that our model estimation is carried out through sampling, not optimization, which may be important in situations with vague prior knowledge such as that encountered in an exploration setting.

P1 l14: "model results": results of the technique introduced here? Could be clearer.

AGREE. This has been changed to "3D model results" to help the reader understand the link to the previous sentence.

P1 l16-18: "The boundaries between geological units are characterized by narrow regions with <95% certainty, which are typically 400–1000 m wide at the Earth's surface" what is the relation between these values and the sampling of gravity data? If I am correct the data sampling of gravity data guaranteed by GSWA is that there is a data point every 400 to 1000 m in the area?

The nominal station spacing for the gravity data is 2.5 km. Interpolation of these data yield a datapoint every ~400 m but this interpolation is not a Bayesian method. Instead, we have used the original 2.5 km spacing to avoid introducing correlations in the interpolation process.

P1 l18: "Beyond âLij4 km depth, the model requires drill hole data". You need to be clearer here. Drillhole data that reache below 4km in the area is not likely in hard rock scenarios, although it might be in oil and gas exploration (basin scenarios). I suspect that you mean that for model cells below 4km the addition of constraints at depth such as drillhole data might help constrain the deeper regions better?

PARTLY AGREE. To avoid confusion in the abstract, we have removed the drill hole constraints. We have left the option of seismic data to be able to constrain models at >4 km depth.

P1 l27: "faults or suture zones": how about unconformities in general?

AGREE. This line has been changed to "...via unconformities or structural discontinuities such as faults or suture zones."

P2 l13: You cite Pakyuz-Charrier et al 2018 but this is a conference abstract. There are two journal papers relating to their MC approach for geological modelling that appeared in 2018. Please replace that reference by the most appropriate one (or both if you want

to be broader) of the following: Pakyuz-Charrier et al. (2018a), (2018b).

AGREE. We apologize for referencing this conference abstract. Both Pakyuz-Charrier papers have now been cited.

P2 l13-14: There have been metrological studies published in recent years that tackle the issue of modelling the uncertainty on geological measurements.

As mentioned above we were aware of Pakyuz-Charrier et al 2018(a,b) and references therein, which deal with the propagation of uncertainties on structural measurements and contact point measurements from drill holes. We were unaware of any previous treatment of lithostratigraphic measurements at the surface, which occasions the new likelihood we derive in this work.

P2 l21-22: "However, there is still a paucity of work in fusing solid Earth geological observations and geophysical data in a Bayesian framework to develop robust 3D geological models". True, but you may need to consider Wellmann et al. (2017), who "...address these shortcomings here with an approach for the integration of structural geological and geophysical data into a framework that explicitly considers model uncertainties [...] in probabilistic programming in a Bayesian framework". Please cite this work. This also relates closely to de la Varga et al. (2018), which you cite earlier in this paragraph. Likewise, Jessell et al. (2010), (2014), (2018) highlight the need for robust geology-geophysics integration. I suggest to cite some of these as they strongly advocates for the kind study presented here.

AGREE. All these papers have now been cited in these two sentences: "...(iv) fusion of structural geology data with geophysical datasets (Wellmann et al., 2018). Despite a clear need for Bayesian fusion of solid Earth geological and geophysical datasets (Jessell et al., 2014; Jessell et al., 2018; Jessell et al., 2010), there is still relatively little work in developing robust 3D geological models, particularly at the local and camp-scale."

[Figure]

P2 l31-33: "One useful addition to the current features of Obsidian would be the integration of geological and geophysical field observations made on the Earth's surface, which are vital for surface and near-surface applications (< 1 km)" this statement sort of contradicts the 1st sentence of this paragraph where you say "The Obsidian software package provides a workflow to fuse disparate geological and geophysical data within a Bayesian framework".

AGREE. We have updated the first sentence to emphasize Obsidian's distinctive value-add, which is its distributed MCMC sampler for 3-D geological models conditioned on geophysical survey data. We then say, "Although previous iterations of the Obsidian software package could not fuse geological field observations made on the Earth's surface with geophysical survey data, relatively little amendment to the program is required to make this possible."

P3 l1-2: "geophysical observations" and "geophysical survey data": how is it not the same thing?

AGREE. Geophysical observations has been deleted. This was an oversight during text editing.

P3 4-14: Maybe say somewhere that exploration undercover has been recognised to be important for the future of mineral exploration with a ref or two. The last sentence of this paragraph could also go in conclusion.

AGREE. Added a final sentence to the first introduction paragraph: "In a future where exploration under cover has been recognized as vitally important for the mineral exploration sector (McFadden et al., 2012), developing geological models with accounted uncertainty is pivotal."

P3 section 2.1: Lead authors Johnson and Sheppard are cited a number of times - consider adding work from someone else.

DISAGREE. These two researchers have worked most thoroughly on the tectonic history of the Capricorn Orogen.

P4 section 2.2: Sambridge and Mosegaard are cited many times here – maybe add some diversity with papers coming from other researchers.

AGREE. This section has now been peppered with many other references.

P4 l14: "single unique" → "unique" is enough.

AGREE. Single has been deleted.

P4 l16: There are also other works you may want to cite when it comes to using infomation derived from geological measurements or modelling directly into geophysical inversion. For instance, Fullagar et al. (2008), Guillen et al. (2008), Scholl et al. (2016) integrate geological information or modelling in their inversion algorithm. Publications relating to works using level-set inversion also rely on geological models (see for example Bijani et al., 2017, and Zheglova et al., 2018, for joint inversion).

AGREE. We have updated this sentence to: "Ways to introduce such constraints include 3D geometry inversion (Fullagar et al., 2008; Guillen et al., 2008), level-set inversions (Bijani et al., 2017; Zheglova et al., 2018) and (cross-)gradient regularization (Giraud et al., 2019; Scholl et al., 2016). However, these techniques are deterministic, yielding a single geological-geophysical inverse model that represents only one scenario."

P4 l15-16: "Ways to introduce such constraints include regularization (Giraud et al., in review) but this technique fails to acknowledge alternative scenarios." This statement in unclear as Giraud et al derive constraints from a collection of geological models, therefore using all realizations from MC sampling of geological model space. But because their inversion is deterministic, they obtain a single geolophysical inverse model, which indeed represent a single scenario. I think that this is what you mean but it needs to be clarified.

AGREE. See previous comment.
P4 l21-30. "The Bayesian framework converts a deterministic model into a probabilistic one by using probability distributions to represent the free parameters rather than using optimal or single-point estimates." To make this clear and unambiguous, you should remind the principle behind Bayesian approaches, when you invoke Bayes' theorem. Solving a problem in a probabilistic way does not necessarily make it Bayesian. In this sentence you may want to stress the fact that you also use the prior distribution and sample the posterior, as it is a major difference with deterministic inversions. The utilisation of the priors is stated in the next sentence but I think that it could be made clearer overall. I would also not cite Oldenburg 2005 here, but perhaps one of A. Tarantola and others' publications which are seminal to many inversion approaches.

AGREE. We have updated the text to include an invocation of Bayes's theorem. We also include descriptions of the Metropolis-Hastings algorithm later on.

P5 l1: I strongly suggest that you add the mathematical foundation of your Bayesian inversion methodology. Just a few of the equations centre to your modelling approach would do and I think be informative to the reader.

AGREE. Equations we have added in response to this comment include: the Metropolis-Hastings criterion, the swap rule for parallel-tempered MCMC, the Crank-Nicholson proposal, explicit derivations of the likelihoods we use, and definitions of the PSRF and Geweke convergence metrics for MCMC sampling.

P5 l10: "convergence can be challenging" rephrase. Something like: "convergence can be difficult to reach"

AGREE. This has been changed.

P5 l13-24: Please add other cites to your citations of works by M. Sambridge and his team's. For instance, they may have brought lots of new ideas and methods to the field but the Metropolis-Hastings algorithm is not by them.

PARTLY AGREE. Both Metropolis et al., 1953 and Hastings, 1970 are already cited in

the next paragraph, where we believe it is most appropriate.

P5 l23: section 2.3. You need to give more information about Obsidian. It is too short and not sufficiently informative. The main reference you use (Schalzo et al in review) is still in review a basic summary of Obsdian should be self-contained in the paper.

AGREE. We have rewritten substantial parts of sections 2 and 3 to provide more specifics relevant to our problem.

P5 l31-32: This sentence is exactly the same as Scalzo et al., in review, beginning of section 2.4. Layers are not discrete if they have smooth boundaries. I suspect that what you try to say is that the outline of a given layer is not angulous (i.e., that its representative functional would be differentiable)? If so this should be made clear.

AGREE. We have updated the text and now specify that the layers are differentiable (as they must be if a square exponential kernel is used).

P6 l2-4: "The layer boundaries are indexed in a strict order of increasing depth in the subsurface but are permitted to cross." The fact that they cross violates basic stratigraphic principles. You need to state explain briefly how you cope with that and how this is dealt with by your algorithm. This sentence is exactly the same as Scalzo et al., in review before they introduce equation 9.

AGREE. This phrasing was unclear. We mean that the ordering constraint among the depth to each layer is not explicit in the parametrization, but is enforced at the stage when the model is discretized for calculation of the potential field forward models. We now state this expressly: "The constraint $z_i(x,y) \leq z_{i+1}(x,y)$ for each layer $i$ is enforced at this stage, allowing layers with no coverage in a particular region to "pinch out" to zero thickness."

P6 l10: "x and y correlation length" you haven't introduced what x and y are, except in Fig. 2. Just say that your RBF is anisotropic and I think that it's enough. The first sentence of this paragraph is very close to the one preceding equation 10 in Scalzo et

al., in review.

AGREE. We now specify that the kernel is anisotropic and that the covariance length matches the lateral grid resolution between control points.

P6 l13-15: please reformulate the last sentence. The parameter alpha and beta haven't been defined.

AGREE. We now defer discussion of these parameters to section 3.4 (on likelihoods) where alpha and beta are defined when they are used.

P6 l18: PTMCMC: this acronym has not been defined yet.

AGREE. This has now been defined in the last paragraph of section 2.2.

P6 l32: "and cross-cuts" you cannot rule out all uncertainty about the fact that it cross-cuts, but you are making the (reasonable) assumption that it does. Please add this information.

AGREE. This has been added.

P7 l1: "ordering of layers". Say "stratigraphy" or "stratigraphic pile".

AGREE. Changed "ordering of layers" to "stratigraphy"

P7 l6-9: typo in line 6. I don't think that a lithology with 3% of occurrence is insignificant. You need to provide more information as to why you do not account for rock units except Halfway Gneiss and Durlarcher Supersuite.

DISAGREE. These lithologies are less significant volumetrically, appearing primarily near the surface, and are also present areas smaller than our model can resolve. We have focused in this work on mapping the boundary at depth between the two major units. More detailed future work will undoubtedly move towards sampling of more detailed models that can resolve finer features.

P7 l14-16: how do your values compare to litterature values, or work reported in the

area or similar settings? If you have only approx. 100 samples to characterize several rock units through mean and standard deviation you can assume that your uncertainty on these parameters is quite high.

These are all the petrophysical values available from the area.

P7 l24-32 and 1st paragraph in P8: I don't think that it is necessary to give detailed information about the processing of the field measurements. You can keep it, but I recommend to put it in Appendix instead of the full text.

DISAGREE. We think this information is particularly important as it highlights the relatively minor pre-processing of the data, which is non-probabilistic. The interpolation vs. original data is also an important point that, in our opinion, needs to be kept in the main text.

P8 l15: consider replacing 'acquired' by 'obtained'.

AGREE. Replaced.

P8 l21: I think that you should replace ">" by plain text words.

AGREE. Replaced with "...more than 100 age and over 500 samples with..."

P9 l2-3: please explain briefly how the shape parameters were obtained.

AGREE. These parameters constitute a prior elicited from our geological expert collaborators and we have added text to explain this.

P9 l9: please number the equations.

AGREE. This has been done

P9 l13-14: the calculation of this integral is not straightforward. If you want to leave it in the manuscript as it is please add a short appendix explaining how it is calculated, and at least provide the appropriate references.

AGREE. As mentioned above, we now include detailed derivations of both the betabinomial likelihood (the reference for the original comment) for the lithostratigraphic observations, and the Student's-t likelihood for the potential-field observations.

P9 l17: please define what 'ID dataset' is.

AGREE. Replaced "field ID dataset" with "field observations dataset"

P9 l25: the 'iGRW' acronym is not used in the rest of the text. Delete.

AGREE. Deleted.

P10 paragraph 1: please add a figure to help the reader to understand how this works. The readership of Solid Earth might not be specialist in MCMC techniques.

AGREE. We now include a new figure (new Figure 3) illustrating the operation of parallel-tempered MCMC, showing trace plots and marginal distributions of each chain operating on an easily visualized example distribution. We hope this will help clarify the algorithm's operation for the reader.

P10 l 13: This information is relevant only if you provide information about the computing resources you used.

AGREE. This section was poorly worded, the true resource use should be measured in CPU-hours and not merely walltime (though arriving at a tractable result in a reasonable walltime is also important). We have revised the text accordingly.

P10 l17: please add in the methodology the definition of the indicators you use to analyse your results.

We have now added more detailed descriptions of the autocorrelation time, the PSRF, and the Geweke score.

P10 l22: I can make an educated guess about what sigma means here, but it needs to be clearly stated.

AGREE. This has been replaced with: "(uncertainties quoted at two standard deviations [2$\sigma$] here and throughout)

P11 l13: the reference to Geweke score should come earlier in the text. The ref given might not be the best. I would cite the following instead: Geweke, Evaluating the accuracy of sampling based approaches to the calculation of posterior moments, 1992, Bayesian Statistics 4, pp. 169-193

AGREE. This reference has been replaced here and added to the first paragraph of section 4.1.

P11 section 4.2. Although it's not perfect as an indicator, I'd also give the root-mean-square error. As said above I do not provide review comments on the results section. From here I go straight to section 5.2.

If by "root-mean-square error" the referee is referring to the residuals of the posterior mean forward-model predictions from the data, this is easy enough to add, and we have done so. P14 l20-21: "So, if higher resolution geophysical surveys and/or geological field observations are acquired, the model can then become more precise". This is not necessary as it is obvious.

AGREE. This has been deleted.

P14 l24-25: "Where such regions are under cover and drilling is required to establish formation contacts, our results also aid in constrainingwhich areas should be drilled first to maximize information gain". This is true as a first approximation but not always valid. You can imagine that adding more information in a certain part of the model may improve greatly a portion of the model that is equally uncertain because it is linked to that first structure in a structural or topologic sense.

AGREE. We have added "could also aid" to allow for the fact that our approximation may not always be true.

3. References Bijani, R., P. G. Lelièvre, C. F. Ponte-Neto, and C. G. Farquharson, 2017, Physical- property-, lithology- and surface-geometry-based joint inversion using

Pareto Multi- Objective Global Optimization: Geophysical Journal International, 209, 730–748. Fullagar, P. âĹijK., G. âĹija. Pears, and B. McMonnies, 2008, Constrained inversion of geologic surfaces - pushing the boundaries: The Leading Edge, 27, 98–105. Guillen, A., P. Calcagno, G. Courrioux, A. Joly, and P. Ledru, 2008, Geological mod- elling from field data and geological knowledge. Part II. Modelling validation using gravity and magnetic data inversion: Physics of the Earth and Planetary Interiors, 171, 158–169. Jessell, M., E. Pakyuz-charrier, M. Lindsay, J. Giraud, and E. de Kemp, 2018, Assess- ing and Mitigating Uncertainty in Three-Dimensional Geologic Models in Contrasting Geologic Scenarios: , 63–74. Jessell, M., L. Aillères, E. De Kemp, M. Lindsay, F. Wellmann, M. Hillier, G. Laurent, T. Carmichael, and R. Martin, 2014, Next Generation Three-Dimensional Geologic Mod- eling and Inversion: SEG Special Publication 18, Chapter 13, 261–272. Jessell, M. W., L. Ailleres, and E. A. de Kemp, 2010, Towards an integrated inversion of geoscientific data: What price of geology? Tectonophysics, 490, 294–306. Kamm, J., I. A. Lundin, M. Bastani, M. Sadeghi, and L. B. Pedersen, 2015, Joint inversion of gravity, magnetic, and petrophysical data âAËŸTËĞA case study from a gabbro intrusion in Boden, Sweden: GEOPHYSICS, 80, B131–B152. de la Varga, M., A. Schaaf, and F. Wellmann, 2018, GemPy 1.0: open-source stochas- tic geological modeling and inversion: Geoscientific Model De- velopment Discussions, 1–50. Pakyuz-Charrier, E., M. Lindsay, V. Ogarko, J. Giraud, and M. Jessell, 2018a, Monte Carlo simulation for uncertainty estimation on structural data in implicit 3-D geological modeling, a guide for disturbance distribution selection and parameterization: Solid Earth, 9, 385–402. Pakyuz-Charrier, E., J. Giraud, V. Ogarko, M. Lindsay, and M. Jessell, 2018b, Drillhole uncertainty propagation for three-dimensional geological modeling using Monte Carlo: Tectonophysics. Scholl, C., J. Neumann, M. D. Watts, and S. Hallinan, 2016, Geologically constrained 2D and 3D airborne EM inversion through cross-gradient regularization and multi-grid efficiency: ASEG-PESA-AIG 2016, 1–6. Sun, J., and Y. Li, 2016, Joint-clustering inversion of gravity and magnetic data applied to the imaging of a gabbro intrusion: SEG Technical Program Expanded Abstracts 2016, 2175–2179. Wellmann, J. F., M. de la

Varga, R. E. Murdie, K. Gessner, and M. Jessell, 2017, Un- certainty estimation for a geological model of the Sandstone greenstone belt, Western Australia – insights from integrated geological and geophysical inversion in a Bayesian inference framework: Geological Society, London, Special Publications, SP453.12. Zheglova, P., P. G. Lelièvre, and C. G. Farquharson, 2018, Multiple level-set joint inver- sion of traveltime and gravity data with application to ore delineation Ÿ ′r: A synthetic study: , 83.  

Please also note the supplement to this comment:
https://www.solid-earth-discuss.net/se-2019-4/se-2019-4-AC1-supplement.zip

---

## Author Comment (AC2) · 3 May 2019

Reviewer #2:

I thank the editor for inviting me to review this paper. Unfortunately, I find this paper quite underwhelming. The paper describes very little that has not already been shown in previous works, nor do the results convincingly reveal new understanding from the region. It is thus difficult to understand what contribution this study makes either to probabilistic methods in geophysics, or geological understanding of the Gascoyne Province. This is compounded by the authors' inadequate review of existing work and failure to place theirs in context with the discipline. The authors make statements

about the “importance” or their results with no justification, and how that their method “is the only technique that provides a range of solutions” which is false. To reiterate, the authors needs to spend more time reviewing the existing literature. One positive is that the manuscript is well-written and structured. I suspect the authors will be able to remedy many of the deficiencies listed here and produce an adequate revision.

We thank the reviewer for their rapid and detailed review of our manuscript. Please find attached a zip file containing comments to all reviewers, a manuscript with track changes and a clean manuscript with all changes incorporated.

I list the major issues directly below, followed by relatively minor comments. Major comments. Valid criticism is made of existing work in the Introduction (P2, L9-14), and refers to how “these approaches still require a significant degree of human decision making into how to fuse disparate geoscientific datasets.” And “these approaches still largely elide the question of how the joint distribution of such parameters is meant to be derived.” This infers the manuscript will then address these important issues, which it barely does. These statements are then followed by another which claims the presented method “will fuse all available constraints in a probabilistically rigorous fashion.” The method doesn’t fuse all available constraints (see discussion, where this is admitted), in fact it only uses a small subset of available data. One major omission is structural and drillhole data, which is used or can be used in all the methods described in the papers cited in this paragraph. These claims are at best poorly made, and at worse false. Pakyuz-Charrier and Giraud both address the issues of how joint distribution of parameters (geophysics, drill holes, petrophysics) are made. A far better justification of these statements needs to be made in order to emphasise the contribution of this paper to advancing this important area of research.

AGREE. Also in light of comments made by reviewer 1, we have modified the second and last introduction paragraph significantly to incorporate the above comments. The structural data would be an excellent addition but, unfortunately, Obsidian is not capable of integrating this data. Future work will involve modifying the Obsidian code but

we felt that the joint inversion of gravity, magnetics, petrophysical and lithostratigraphic data was sufficient for this contribution. Drill hole data is scant in our study area, with only a few ~10 m deep holes located in the SW corner.

Comparing models results with maps. If the maps were made using interpretations from geophysics, then the model, which is based on geophysics, matching the map is not surprising, and expected, and thus not an adequate validation exercise. Please better justify the validation method.

PARTLY AGREE. The geological maps were primarily derived from geological mapping. There is some significant regolith cover in the southern portion of the region but the majority of the region is able to be mapped without the need for geophysical input.

How is the geological model built? Figure 2 implies five units are modelled, and then it's revealed deep into the discussion that only two were modelled. Differentiating between two geological units is not that exciting, not useful, especially at the scale of the study, so the authors need to show better justification as to how this method is novel, and worthy of publication.

One point of novelty we believe we have underemphasized in the previous version of our manuscript is that while the parametrization of our 3-D geological model is not very sophisticated, it incorporates very little prior information about the region in question, in comparison to more detailed parametrizations that start from a detailed geological survey. This may make our approach useful in a greenfields exploration context, perhaps to produce initial 3-D models that can be refined spatially in separate runs as more data become available. The limitations on what kinds of geological features can be represented by our choice of parametrization is something we hope to address in future work.

Two important papers that are not referred to and are very relevant to this work are: Wellmann, J. F., M. de la Varga, R. E. Murdie, K. Gessner and M. Jessell (2018). "Uncertainty estimation for a geological model of the Sandstone greenstone belt, Western

Australia – insights from integrated geological and geophysical inversion in a Bayesian inference framework." Geological Society, London, Special Publications 453(1): 41-56. Guillen, A., P. Calcagno, G. Courrioux, A. Joly and P. Ledru (2008). "Geological modelling from field data and geological knowledge: Part II. Modelling validation using gravity and magnetic data inversion." Physics of the Earth and Planetary Interiors 171(1-4): 158-169.

Both these works describe methods similar to that being described here and deserve a good review in this paper. In particular Wellman et al. 2017 presents a Bayesian framework for geophysics that authors would benefit from during their review.

AGREE. Reviewer 1 also noted that these papers were omitted. Both papers have now been cited in the text, with particular reference to the structural measurement integration covered in Wellmann, et al., 2018.

No figure shows any 3D model, either the initial, or geophysically constrained geological version, nor the inverted geophysical volume. This is a critical thing to show to the readers of Solid Earth, most of whom are geoscientists. How can we appreciate your endeavours without seeing the results, especially when "3D geological models" is in the title?

DISAGREE. Figure 11 shows two 3D models.

Downsampling of "geological" (really lithostratigraphic) observations. You have detailed "petrographic, geochemical and geochronological knowledge obtained on a subset of WAROX data" which would surely give far higher lithological resolution that the five bulk units that make your model (legend of fig. 5). How did you downsample these observations into the five major groups? As you hint, there is significant uncertainty, not just in correctly identifying the correct rock unit (though with the data you have this source of error should be reduced, but not irreducible). This alertoric uncertainty is inadequately addressed in P9, L19. How did you determine the error in these observations? How was it translated into a Beta distribution? There is also the loss of information from the

process of downsampling – i.e. epistemic uncertainty (which is reducible). You refer to section 3.4 in this matter, but section 3.4 barely describes this in a geological context. Other issues with section 3.4 exist. . . next paragraph.

We are not sure what the reviewer means by this.

Section 3.4 needs significant work. It is quite disjointed from the previous section. For example, how are the survey forward models determined? Why alpha = 1 and Beta = 2? There is no effort to make the text link with the previous sections, explain the importance of a Beta distribution to a Bayesian framework, nor appropriate translation of known uncertainties within a geological context or even related to widely understood sources of uncertainty in geoscientific data. In its current form this section is incomprehensible.

AGREE. We have completely rewritten this section in the hopes of improving clarity, explaining that the alpha and beta parameters (for the inverse-gamma and beta priors on sensor noise) are elicited from experts, and that they are fairly vague priors.

The results are not presented well. Section 4.2 Residuals from forward models: Statistics are presented without any indication as to whether they are acceptable (e.g. “Aeromagnetic residuals display an approximately Gaussian distribution of 0 +358 –31725 nT ($2\sigma$, 21% of the total magnetic range” – so what?) or even higher or lower than expected.

PARTLY AGREE. Results should be as transparent as possible with as little as possible interpretation. Stating that something is acceptable or unacceptable is subjective and is appropriate in the discussion, not the results. The implications of the distribution and ranges are already discussed in section 5.1. Nevertheless, we agree that the results could be better streamlined and have made comparisons between the total range of each of the sensors to the petrophysical data to help the reader understand this link.

Section 4.3 Probability density of layer locations. Text associated with figure 9 states

that rock observations near the contact between the Halfway Gneiss and Durlacher Supersuite are misclassified. None of this is very surprising given it's a contact which any geologist knows are hard to define. But the relevance of this finding is difficult to discern given the method for building the model isn't described anywhere, the cell sizes of the model are not given (see comments below), nor how the contact was defined in the first place. All it infers (given no other information) is that Obsidian doesn't manage to determine the geometry of this contact well. This may not be true, but none of the other results presented show that Obsidian has done a good job in this regard. This is not helped with the lack of description for geological model construction.

AGREE. We have clarified these aspects in our revised text. The primary aim of our work is to provide a quantitative framework for the uncertainty in the location of the contact in our model, which makes the statement that contacts are "hard to define" more precise. Unlike more detailed parametric models, we arrive at our posterior probability distribution for the contact geometry starting only from a minimum feasible resolution – set by the discretization and the correlation scale for the interface geometry – and the knowledge or strong prior belief that a contact exists in the modeled volume. We also set out additional details of the geological model in revisions to sections 2.3 and 3.1, describing the grid of control points, the covariance kernel for the interpolating surface defining the contact, and the discretization resolution.

You state that results show the Durlacher Supersuite to be in two "domains". This isn't surprising given it is a Supersuite, and by definition made up of multiple suites, which can be defined as domains. This interpretation is also not supported by any geological data, nor is the importance of this made apparent during the introduction, discussion or conclusion. The results are described as being far more successful than they really are.

DISAGREE. The petrophysical data, which is inherently tied to the geophysical data, shows there is no spatially-controlled differences between the different domains of the Durlahcer Supersuite.

Page 14, line 13: “Highly similar” – not really. There are a quite few differences, plus you have only shown the probabilities of two units, when the 3d model was built using 5 units (maybe? Again, describe how the model was built). What about the other three units? The assessment of this method is thus inadequate. As such much of the discussion unconvincing, especially when two select slices of the probabilistic model are shown.

AGREE. We agree that parts of the methods section were ambiguous and we have now made these more transparent. Only the 2 most voluminous units were modelled as the other 3 units are volumetrically and areally minor. We have also changed some of the figures of Figure 11 (3D model) to make it clearer where regions of uncertainty are.

You admit that structural data is not used in the discussion (P14, L29). This needs to be stated clearly in the method (where a description of model construction is required – see previous comments) and makes earlier comments criticizing previous work disingenuous (see major comments). It is self-evident that structural data is very useful for geological models. The use of structural data is shown in other methods that have been around for almost a decade (see uncertainty work by Wellmann, de la Varga, Bond, Lark, Lindsay, Jessell), or general modelling (see Calcagno paper). Why can you not do this? The same can be said for drillhole data, which other methods also use. The main problem is that both structural and drill hole data from the area is publically available, but not used. So it appears that Obsidian, or the described method cannot use these data presently, otherwise they would have done so. Other methods (as cited earlier) can use both structural, seismic and drillhole. How do the authors then justify this method as novel, or one people should adopt given is has severe limitations to inputs? Simply being Bayesian is not enough, especially as Bayesian methods are well suited to integration of different data types. Other Bayesian techniques have been proposed (de la Varga, Wellmann). This aspect needs a much fuller justification and discussion.

PARTLY AGREE. This has now been made upfront in the final section of the introduction instead of in the methods: “There are a few datasets available in the region that are not utilized in our model. (1) There are only a few ∼10 m-deep drill holes in the southwestern corner, so drill hole data is omitted as it does not add further detail than surface observations provide. With a lack of drill hole data, our contribution is able to address the impact of solely surficial geological data on the model accuracy. Applications such as greenfields mineral exploration or tectonic analysis of hard rock terranes without drillholes would benefit from this understanding. (2) Our study excludes the use of structural geological data because other workers have recently focussed on this problem (e.g., Pakyuz-Charrier et al., 2018b; Wellmann et al., 2018) and because Obsidian cannot yet incorporate structural data. (3) The single 2D active seismic line immediately to the west of our model (Fig. 1) is not utilized in a Bayesian framework because the vast majority of hard rock terranes do not have seismic data coverage.”

Minor Comments Page 3, line 7: define “data-rich”. Rich in diversity or coverage, both? Quantify this richness.

AGREE. This has been changed to: “We demonstrate the validity of our techniques by building models of a 13.5 × 13.5 km subsection of the Gascoyne Province, Western Australia (Fig. 1), that is rich in data diversity and coverage”

Page 4, line 11: technically measurements of gravitational acceleration and magnetic field strength

AGREE. This has been changed.

Page 4, line 16 – Be clear about the shortcomings of other work. Giraud et al. in review does acknowledge alternative geophysical, petrophysical and geological sce- narios. Are you referring to alternative forms of parameterisation for regularization?

AGREE. Also in line with reviewer 1’s comments, we have changed this to:

“Ways to introduce such constraints include 3D geometry inversion (Fullagar et al.,

2008; Guillen et al., 2008), level-set inversions (Bijani et al., 2017; Zheglova et al., 2018) and (cross-)gradient regularization (Giraud et al., 2019; Scholl et al., 2016). However, these techniques are deterministic, yielding a single geological-geophysical inverse model that represents only one scenario."

Page 4, line 27: it is unclear what you mean by "geophysical processes". Are you talking about how well the models represent the geology?

AGREE. It was unclear to us exactly what we meant with "geophysical processes" as well. Apologies about that. We have now changed the text to:

Page 6, line 18: "PTMCMC" define your acronyms before using them.

AGREE. This has now been defined in the last paragraph of section 2.2.

Subheading 3.1. "World" is an expansive term that infers all parameters, data, models, inferences, assumptions are under consideration in the following paragraph, which isn't true. "3D geological model parameterization" is more specific and less confusing. The same applies to all references of "world" models. This is important as you use and describe more than one model through the manuscript, including statistical, geophysical and conceptual are present as well.

AGREE. The title of subsection 3.1 has been changed to "3D geological model parameterization" and any "world" text replaced by "3D geological model" in the rest of the manuscript.

Page 6 line 30: should be "magnetic susceptibility and density data"

AGREE. This has been changed.

Section 3.2: You need to show where these petrophysical data were acquired on a map (Fig. 2?). Presumably the petrophysical data locations will be different to the "surface observations" shown in Fig. 2. Given the caption describes them as geological surface observations

AGREE. These have now been shown in Fig. 1.

Page 7, line 21. Please describe what Bayesian “fusion” is, or just say the data were input to the Obsidian framework.

AGREE. This was confusing and has now been updated to: “These types of geophysical surveys were already available for incorporation in the Obsidian framework”.

Page 7, line 31: explain the source of these “correlations”, what they are correlated with, and why this could be a problem. You do this later with the gravity data (P8,L10), so move that explanation here. But you still need to better explain the source of the biases and how they produce incorrect results in context of the Bayesian methods you describe earlier.

AGREE. The section following the gravity data has now been moved up.

Page 9, Line 23 PT-MCMC or PTMCMC (as P6,L18)

AGREE. All is now referred to as PT-MCMC.

Page 11, Line 18 Discussion of large Gweke scores

AGREE. We have now discussed this in the discussion (not in the results)

Page 11, Line 21 Figure 8 needs to show the measured interpolated image with the forward models for easy comparison, rather than forcing the reader to switch between figures on different pages.

Figures 8a and d already do this as stated in the caption: “Modelled mean contours of (a) Bouguer anomaly and (b) magnetic intensity compared to interpolated mean colored data.”

The reader is also referred to figure 2 when describing magnetic lineaments, but figure 2 is a geological map. Are the authors assuming the NW strike of the geology will also produce NW striking magnetic lineaments? This is a reasonable interpretation, but the

authors need to first make that interpretation for the statement in line Page 11, Line 22.

AGREE. This was poorly worded. It now reads: "Aeromagnetic models effectively identify the NW–SE strike of magnetic lineaments in the northern half of the modelled volume (Fig. 8) that would be predicted from geological maps (Fig. 2)."

Page 11, Line 25 – okay, but so what?

AGREE. This paragraph has been overhauled to make it clearer for the reader. We also discuss the implications of these ranges more fully in section 5.1:

"Aeromagnetic residuals display an approximately Gaussian distribution with a mean of 0 nT (i.e., equivalent to measured aeromagnetic data) and a range of "+358" Âę"–317" nT (2$\sigma$), which covers approximately 21% of the total magnetic range (Fig. 8). The $\sim$21% residual standard deviation is comparable to the standard deviation range of magnetic susceptibility values (Fig. 4). Only one region in the northwestern portion of the map has significantly higher magnetic field strength than modelled (Fig. 8)."

Page 11, Figure 8 caption. Labelling of figure parts appears to be incorrect. b) shows units in mGal, so not magnetic intensity, c) shows a histogram, not the modelled contours

AGREE. We thank the reviewer for identifying this mistake. The lettering in the legend has been corrected.

Page 12, Line 14. Distances have no meaning without telling us what the model cell sizes are first. How many cells does 300-1000m represent?

AGREE. The model cell sizes are 500 m, as now stated in section 3.1.

Page 12, Line 14. "The ellipsoidal Durlacher Supersuite inlier is heterogeneously constrained." What does this mean and why is it relevant?

AGREE. This was difficult to follow. We have changed it to: "The uncertainty on the boundary between the ellipsoidal Durlacher Supersuite inlier and the Halfway Gneiss

is constrained differently in different parts of the model." This follows better to the following sentences.

Page 13, Line 3 "a function of a long-wavelength (i.e. deep) gravity response" Careful here. A long wavelength is not always deep. It can be laterally extensive but shallow.

AGREE. To avoid ambiguity, this sentence has been changed to: "This is primarily a function of a long-wavelength gravity response that is probably attained from the deep subsurface (Johnson et al., 2013)."

Page 13, Line 6. I would think more petrophysical data from "other geological units" (see previous sentence) would be more useful to define the lithostratigraphic diversity than marginally tightening the standard deviation of the Halfway Gneiss and Durlacher SS units.

PARTLY AGREE. More petrophysical data from all lithostratigraphic units in the Gascoyne Province would be beneficial (see Fig. 1). There are only a few on the periphery of our model currently. In any case, more petrophysical data from the two volumetrically-major units in our modelled region would have improved the local-scale model, which is an important point for mineral explorers.

Page 13, Line 11-20 Annotate the figures with the various features being described here (Chalba SZ, Durlacher SS 'spur' and 'sliver' etc.)

AGREE. This has been added to Fig. 10a. We also considered adding it to Fig. 2a, but Fig. 2a is already relatively cluttered.

Page 13, line 19-20. Tells me the initial 3D model is wrong.

While we acknowledge that our parametrization is limited, we should also point out that there is no "initial" 3-D model in the way the reviewer would understand it. Instead we use some rather restrictive assumptions about the form a contact would take (the square exponential covariance for a Gaussian process defining the interface, and the control point geometry) but an essentially random initial guess for its geometry. While

most methods thus start close to a known or believed model (or combination of parameters), our sampling process explores all models of the specified parametric form that are compatible with the data. The chosen parametric form has substantial limitations discussed in our text, but appears to be good enough to capture the overall contact geometry on scales larger than the discretization resolution. Page 13, line 31. "Importantly, our method is the only technique that provides a range of solutions and quantitatively accounts for all the input assumptions" This grandiose statement needs far more justification. I actually think this should be removed entirely, given the technique is poorly described in the first place.

AGREE. We have removed this statement.

Page 14, line 1-9. Figure 10d? Over-interpreted results. "Definitively separated" Plus given the sections only extend to 4km, how can you be sure the Durlacher remains separated beyond that? Figure 10 d looks like the Halfway Gneiss only extends as far as 4km depth (which is also probably a function of the model volume parameters? Also needs discussing). You then state correctly in later lines (lines 4-5) that "it was difficult to know whether this spur of Halfway Gneiss between the two Durlacher Supersuite domains continued at depth or was truncated in the near subsurface". Hardly definitive!

AGREE. The earlier statement is misleading and has been changed to: "…is separated into two domains at the surface and shallow subsurface"

"This important contribution shows that small geological volumes on the scale of a few km can be resolved accurately and will be important when this modelling output is up-scaled to larger regions." Small volumes can be detected given appropriate geophysical data resolution and corresponding model parameters. Small volumes have also been detected by many other methods which I suggest you spend some time reviewing (Li and Oldenburg papers, Peter Fullagar, Guillen, etc etc) so you realise this is not a world first. Upscaling models to larger regions is also commonplace. If you are to make this kind of statement, please explain how upscaling should be done.

AGREE. We have removed reference to upscaling here. We talk about upscaling in more detail in section 5.2, paragraph 3.

Page 14, line 29. How are drill holes going to help Bayesian methods >4km when they rarely extend beyond 400m?

AGREE. We have clarified this to “petroleum wells”, which do go down to as deep as ∼5 km.

Page 15, Line 22. Interesting concept, and I agree should be done, but expand on how this would be achieved?

PARTLY AGREE. At this stage, we’re not sure how it should be done yet.

Please also note the supplement to this comment:
https://www.solid-earth-discuss.net/se-2019-4/se-2019-4-AC2-supplement.zip

---

## Author Comment (AC3) · 3 May 2019

Reviewer #3:

The manuscript entitled "Bayesian geological and geophysical data fusion for the construction and uncertainty quantification of 3D geological models" presents an interesting approach for integrating geophysical and geological observations into a probabilistic modelling approach of subsurface. Joining the gap between geological and geophysical inversion of subsurface has been a long-standing problematics and this study takes one step toward this objective. Being from the geological side of this problematic my main observations are that the geological description of the subsurface that

is used in this study remains relatively simple (probably for sake of parsimony, as required by inversion problems). I would not place this remark as a criticism for this study but rather as an acknowledgment that further research is needed for improving the parameterisation of geological models. This study has the merit of showing the interest of coupling geophysical and geological inversion. Therefore, I find it valuable for the community and I would recommend accepting it for publication provided some minor revisions.

We thank the reviewer for their detailed review of our manuscript. Please find attached a zip file containing comments to all reviewers, a manuscript with track changes and a clean manuscript with all changes incorporated.

2 Specific comments: The parameterisation consist of a Gaussian process with an RBF interpolation of depth map of the geological contacts. You should better discuss the limitations of this approach on the geometry of the contacts. At least it cannot reproduce multiz structured, but it would also have limitations for vertical or sub vertical contacts. As you mentioned in the introduction Obsidian was designed for basins, ie. With layers being roughly horizontal. Isn't it limiting the sampling and more generally the applicability of the method to other geological regions? The description of this geological parameterisation was apparently supposed to be supported by fig. 3 but several reference to this figure in page 6 line 1 and 10 are apparently not pointing at the right thing. The user-defined control points of the Gaussian process are not show as expected. In addition, it would be interesting to show the prior distribution for the parameters describing the depth of the geological interface, which has apparently been omitted. AGREE. We agree that the 2.5-D approach used by Obsidian has limitations when applied to more complex geologies, which we now highlight in more detail in the discussion. We anticipate that many of these insights will carry over to more complex 3-D modeling methods such as the implicit surface approach used by GemPy (de la Varga et al, 2018); future work will combine advanced sampling methods like Obsidian's with sophisticated parametrizations of geological structure like GemPy's. We

have also included mention of the prior distribution for the depth to the geological contact, which was very permissive — a Gaussian with standard deviation 5 km at each control point. Thus while the 2.5-D parametrization with RBF kernel for the interface is restrictive, any further assumptions about depth to contact were by design uninformative. This is unrealistic when trying to reproduce a well-studied area in detail, but could be useful when formulating initial models in areas about which little is known." You chose to ignore some lithologies based on their smaller cover of the surface. What is the mag sus and density of these formations? They are ignored (because barely seen on surface) but are they going to affect the magnetic and gravity field?

AGREE. This was poorly worded before. We have now added to section 3.1: "Only the two volumetrically-major units are modelled in this study as the other units appear primarily near the surface (Johnson et al., 2011), and are also present only in areas smaller than our model can resolve (see next paragraph). Resolving finer-scale features is out of the scope of this contribution."

I think you should clarify the way you introduce and discuss you probabilistic approach of the lithological observations. Unless there are arguments for taking particular care with these observations, they seem to me to be rather hard constraints as compared to gravity and magnetic responses. Of course, observations could be misinterpreted, but unless the two discussed units are very similar, you would not need chemical analysis or dating to assign them to one or the other group. On the other hand, gravity and magnetic field are by nature ambiguous.

AGREE. We have overhauled our methods section to make this significantly clearer.

Why are the high probability areas that outcrop in the middle of the modelled region so different between fig 9b and 10a? On 9b it looks roughly circular whereas it has a crescent shape on 10a.

This is just a distortion in the 3D model. The actual modelled area is circular.

Please refer to the annotated manuscript for more detailed comments and corrections. We have made minor changes as recommended by the reviewer's attachment.

Please also note the supplement to this comment: https://www.solid-earth-discuss.net/se-2019-4/se-2019-4-AC3-supplement.zip